



# A robust gap-filling approach for ESA CCI soil moisture by integrating satellite observations, model-driven knowledge and spatiotemporal machine learning

Kai Liu[1], Xueke Li[2], Shudong Wang[1], Hongyan Zhang[1]

[1]Aerospace Information Research Institute, Chinese Academy of Sciences, Beijing 100094, China.
[2]Institute at Brown for Environment and Society, Brown University, Providence, RI, 02912, USA

*Correspondence to*: Shudong Wang(wangsd@aricas.ac.cn)

**Abstract.** Soil moisture (SM) is a critical component of the water cycle and a key ecological process connecting the soil-vegetation-atmosphere system. Spatiotemporally continuous SM is increasingly demanded for ecological and hydrological

research fields. Satellite remote sensing has opened opportunities for mapping SM. Nevertheless, the continuity of SM imagery is hampered by data gaps resulting from inadequate satellite coverage and radio frequency interference. In light of this, we propose a new gap-filling approach to reconstruct daily SM time series using the European Space Agency's Climate Change Initiative (ESA CCI). The developed approach integrates satellite observations, model-driven knowledge and machine learning algorithm that leverages both spatial and temporal domains. Taking SM in China as an example, we show

high accuracy of the reconstructed SM when validated with multiple sets of in situ measurements, with a root mean square error (RMSE) and mean absolute error (MAE) of 0.09-0.14 and 0.07-0.13 cm$^3$/cm$^3$, respectively. Further evaluation with a 10 fold cross validation reveals a median value of the coefficient of determination ($R^2$), RMSE, and MAE of 0.56, 0.025 cm$^3$/cm$^3$ and 0.019 cm$^3$/cm$^3$, respectively. The reconstructive performance is noticeably reduced when excluding an explanatory variable while the rest remains unchanged, as well as when removing the spatiotemporal domain strategy and

the residual calibration procedure, respectively. Compared to that using satellite-derived diurnal temperature range (DTR), reconstructed SMs using bias-corrected model-derived DTRs exhibit acceptable accuracies and higher spatial coverage. Applying our gap-filling approach to long-term SM data sets (2005-2015), we show a promising result with a $R^2$ of 0.72. A more accurate trend is achieved relative to that of the original CCI SM when assessed with in situ measurements (0.45 versus 0.23 in terms of $R^2$). Our findings indicate the feasibility of integrating satellite observations, model-driven knowledge, and

spatiotemporal machine learning for filling gaps in SM time series over short and long time scales, providing a potential avenue for applications to similar studies.

## 1. Introduction

As an essential component of land-atmosphere interactions, soil moisture (SM) substantially impacts the energy, water, and carbon cycles. It plays an important role in hydrological, environmental, and agricultural applications such as


evapotranspiration estimation (Detto et al., 2006), drought assessment (Wang et al., 2011), and flood forecasting (Wanders et al., 2014). SM has been declared by the Global Climate Observing System (GCOS) and United Nations Framework Convention on Climate Change (UNFCCC) as one of the 50 vital variables in terrestrial domains. Spatially and temporally continuous daily all-weather SM facilitates understanding of ecological and hydrological processes, and a reliable SM dataset is urgently demanded.

There are a variety of ways of collecting SM. In situ measurements can capture the temporal variability of SM at the station scale. Plenty of in-situ monitoring networks have been installed regionally, nationally and globally, e.g., the crop growth and farmland SM database in China, the North American Soil Moisture Database in North America, and the International Soil Moisture Network (ISMN) (Schaake et al., 2004; Dorigo et al., 2011). Nevertheless, owing to the limited ground stations, it is challenging to obtain spatially continuous SM measurements across large-scale regions. In addition to ground-based

observations, SM can be simulated with various numerical models. The Global Land Data Assimilation System (GLDAS) and European Centre for Medium-Range Weather Forecasts (ECMWF) fifth-generation global atmospheric reanalysis (ERA5) can model the soil moisture values that have sufficient spatial coverage (Chen et al., 2013; Reichle et al., 2011). However, these model simulated dataset tends to be sensitive to the uncertainties relating to model structure, forcing, and parameterization (Prihodko et al., 2008).

Satellite observation has been considered as one powerful technique for retrieving surface SM especially accompanying the increasing improvement of sensor technology (Crow et al., 2012). Some SM-dedicated satellites, e.g., the Advanced Microwave Scanning Radiometer-Earth Observation System (AMSR-E), and Advanced Scatterometer (ASCAT) have used the higher C-band and X-band microwave frequencies to collect SM signals (Paloscia et al., 2001). However, these satellite systems are less subject to atmospheric variability and vegetation coverage. Apart from this, some observation sensors are

installed with the lower L-band radiometers, such as the Soil Moisture and Ocean Salinity (SMOS) (Kerr et al., 2001) and Soil Moisture Active and Passive (SMAP) (Entekhabi et al., 2010). These observation systems have exhibited great potential in collecting SM sources due to the strong capacity of L-band in penetrating vegetation. A case worth noting is that the Climate Change Initiative of the European Space Agency (ESA CCI) has generated one set of global SM dataset (Gruber et al., 2019; Dorigo et al., 2017). This CCI SM product blends a series of SM products from active passive microwave satellite

sensors, enabling it one complete and consistent observational SM record. Despite some uncertainties, earlier studies have revealed good accordance between the CCI SM and the in situ measurements over different regions (Dorigo et al., 2015).

Although the active and passive microwave satellite sensors can depict soil moisture characteristics across large scales, the gap issues still exist in these satellite-based SM products. This is related to a variety of factors, such as the radio-frequency interference and orbit changes of satellite sensors. Considerable efforts have been dedicated to filling the missing values in

the satellite-derived SM dataset. Traditional interpolation approaches are applied to fill gaps relying on the spatial or temporal patterns of the target variable, such as the inverse distance weighting and cokriging (Yao et al., 2013; Ford and Quiring, 2014). Some other studies focus on statistical methods that mainly depend on the statistical and physical relationships between target variables and explanatory variables (Leng et al., 2017). Machine learning strategies have been



recently introduced in gap-filling the satellite dataset (Zhang et al., 2021c; Zhang et al., 2021b). These methods have a strong
capacity in depicting complex relationships of target variables and explanatory variables. Compared to statistical-based
models, the machine learning models may be more flexible and robust especially regarding the complex scenes and extended
coverage (Reichstein et al., 2019).

Most current SM gap-filling studies typically rely on explanatory variables that are required in describing SM dynamics. The
common explanatory variables include satellite-derived vegetation index (e.g., normalized difference vegetation index
(NDVI) and enhanced vegetation index (EVI)), surface albedo and land surface temperature (LST), as well a variety of
climatic and geographical factors have been employed in these studies (Almendra-Martín et al., 2021; Cui et al., 2019; Jing
et al., 2018). Nevertheless, most of these variables are less suitable in heterogeneous regions and extended coverage,
although they are suitable for regional areas. For example, previous studies (Song et al., 2021; Liu et al., 2020b) illustrated
that studies focusing on NDVI and LST tend to achieve better performance in delineating SM in arid and semi-arid regions,
but produce unsatisfactory performance in humid areas. Moreover, these satellite-derived variables (e.g., optical and thermal
infrared parameters) are likely to be impacted by cloud conditions. Accordingly, researchers have attempted to explore
effective information for promoting model establishment and application. Some studies use the feature transform approaches
to extract distinct signals for driving models. Principal component analysis (PCA) and wavelet decomposition have been
employed to reconstruct SM and other satellite-based parameters (Uebbing et al., 2017; Almendra-Martín et al., 2021).
Despite pretty good model performance achieved in the humid and semi-arid region (Zhang et al., 2016; Almendra-Martín et
al., 2021), some studies found that there is no substantial improvement in model performance in cropland of semi-humid
region when using the PCA (Wang et al., 2020). Some other studies have focused on the distinct dataset source for gap-
filling models. Soil moisture from GLDAS, ERA5, China Meteorological Administration Land Data Assimilation System
(CLDAS) and Fengyun Microwave Radiation Imager is considered (Long et al., 2019; Cui et al., 2020). The gap-filling
models integrating these unique dataset sources are able to describe SM dynamics, but uncertainties are still observed in the
humid regions and freezing-thaw areas (Song et al., 2021; Cui et al., 2019). Overall, the availability of explanatory variables
in contributing SM reconstructing models is inadequately explored, which is especially critical for machine learning gap-
filling models that are sensitive to the structure of input sequences (Mao et al., 2019).

Although earlier studies have focused on completing the SM dataset, most of them partially aim at the specific case of
satellite observations but less consider the large continental region. Almendra-Martín et al. (2021) and Liu et al. (2020b)
applied reconstruction algorithms to the CCI SM product in regional Europe and Oklahoma, USA, respectively, and Cui et
al. (2019) continuously promote this approach in the Tibetan Plateau. These models rely on machine learning algorithms and
a variety of satellite-based variables. Furthermore, studies aimed at the challenging case of time series SM dataset at the
daily scale are insufficiently implemented (Zhang et al., 2021c; Long et al., 2019), which is fundamental to explore the SM
dynamics and quantify its impacts and contribution on climate change and water cycle.

This study proposes a robust gap-filling methodology for reconstructing a spatially continuous daily ESA CCI SM, primarily
based on satellite observations, model-driven knowledge and one spatiotemporal random forest algorithm. Our model is



applied to continental China which has sufficient landscape variability and climatic conditions. To be specific, the feasibility and merit of the developed model are demonstrated by 1) evaluating the gap-filled results with the in-situ measurements and

the holdout cross-validation, and comparing against those of other models; and 2) discussing the model uncertainty in terms of the filtered explanatory variables, and extending the proposed model to one long-term period.

## 2. Study region and material

### 2.1 Study region

China is located from 3°51′N to 53°33′N and from 73°33′E to 135°05′E, covering an area of approximately $9.6 \times 10^6$ km$^6$

(Fig. 1). A variety of terrain types are presented across China, including the plain, basin, plateau, mountain and hill. These diverse terrains inevitably result in noticeable spatial differences in precipitation and temperature, accompanying the elevation decreasing from west to east. Seven climate zones can be identified in China, including arid, semi-arid, arid/semi-wet, wet/semi-arid, wet, moist, and over-wet climates. The identification of this zoning system is based on a China's humidity index map produced by the National Earth System Science Data Center, National Science & Technology

Infrastructure of China (http://www.geodata.cn).

In addition to the whole regions of China, we also chose two local regions for model uncertainty analysis (Fig. 1). One region is focused on northern China (NC) which is mostly occupied by arid and semi-arid areas, while the other region is focused on southern China (SC) that is occupied by wet areas.

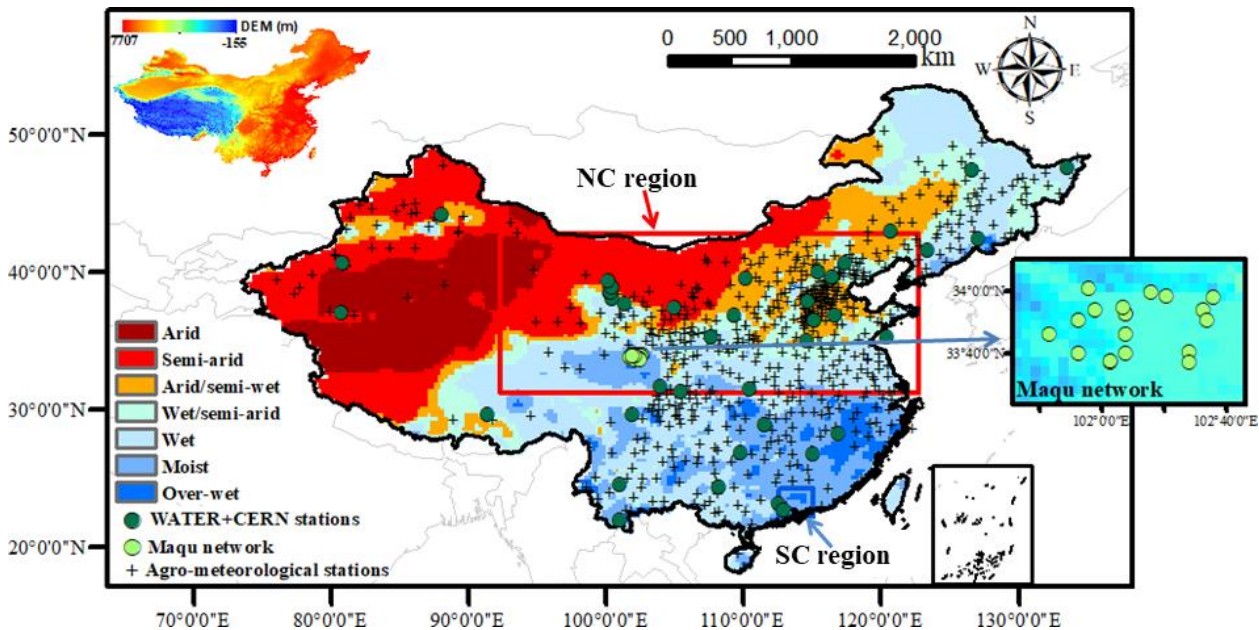





**Figure 1: The study region and the selected in situ soil moisture sites. The figure in the upper-left corner delineates the DEM information. The detailed distribution of dense in situ measurements in the Maqu network is shown in the figure on the far right. Two regional areas for uncertainty analysis (i.e., northern China (NC) and southern China (SC)) are delineated with the rectangle.**

**2.1 Material**

The dataset used mainly includes, i) satellite product, reanalysis dataset, and land surface model products for the model

establishment, ii) in situ measurements for model validation, and iii) reanalysis dataset and land surface model dataset for model performance analysis. Table 1 (and Table S1) provides the information of the used dataset. Details about these datasets are described in the following sections.

**Table 1. Summary of the dataset used for the proposed model. Other dataset for the preliminary analysis but not the final utilization of the model is exhibited in supplementary Table S1.**

| Aims | Variables | Source | Resolution (spatial/temporal) |
|---|---|---|---|
| | Soil moisture | ESA CCI | 0.25°/daily |
| | Surface albedo | MCD43C3 | 0.05°/16 day |
| | NDVI | MOD13C1, MYD13C1 | 0.05°/16 day |
| | Land surface temperature (LST) | MYD11C1 | 1km/instantaneous |
| | Precipitation | Chinese regional ground meteorological dataset | 0.1°/3 hourly |
| | Potential evapotranspiration (PET) | GLEAM | 0.25°/daily |
| | Soil moisture | ERA5 | 0.25°/hourly |
| | Land cover classification | MCD12Q1 | 500/annual |
| Model establishment | Digital Elevation Model (DEM) | SRTM | 90m |
| Model analysis | Surface temperature | Noah simulations from previous work | 1km/3 hourly |
| | Surface temperature | ERA5 | 0.25°/hourly |
| | Surface temperature | GLDAS | 0.25°/3-hourly |
| | Soil moisture | GLDAS | 0.25°/3-hourly |
| | Soil moisture | GLEAM | 0.25°/daily |

**2.1.1 Satellite dataset**

The ESA CCI SM dataset is provided by the Climate Change Initiative program of the European Space Agency. This product is primarily composed of three types of daily dataset sources, i.e., active, passive, and active-passive combined microwave products (Dorigo et al., 2017). Despite the wide spatiotemporal coverage of CCI SM, the data gap remains a major challenge that hampers its further application. Here we select the daily combined microwave products version 4.5,

with a spatial resolution of 0.25°. The inconsistent data in the CCI combined SM is filtered using the quality flag variable.



A variety of Moderate Resolution Imaging Spectroradiometer (MODIS) products are collected, including the 0.05° daily LST (MYD11C1), the 0.05° 16-day composite albedo (MCD43C3), the 0.05° 16-day composite vegetation indices, i.e., normalized difference vegetation index (NDVI) and enhanced vegetation index (EVI), and the 8-day composite 500-meter leaf area index (LAI) (MCD15A2H). All these datasets are collected at MODIS 6 collection. We calculate the diurnal

temperature range (DTR) by subtracting the night LST from daytime LST. The NDVI and EVI are averagely obtained from the two products: MOD13C1 and MYD13C1. All the selected products are screened out using the quality variables to maintain only the available pixels with good quality. We also collect the 0.05° annual land cover product (MCD12Q1) for quality control of CCI SM.

Topography may be related to the spatial distribution of soil moisture. We use the Digital Elevation Model (DEM) dataset

provided by NASA's Shuttle Radar Topography Mission (SRTM) to retrieve several relevant topographic metrics. SRTM dataset has been extensively employed (Van Zyl, 2001), and it has an original spatial resolution of 90 m. The slope, aspect, and the topographic position index (TPI) (Guisan et al., 1999) are used. The TPI is calculated by subtracting the focal grid elevation with and the mean elevation of the eight surrounding grids.

Considering the low accuracy of satellite SM for snow-covered pixels, the pixel that has both daytime LST lower than 0 °C

and the albedo higher than 0.3 are removed (Cui et al., 2020). We also remove the pixel that occupies more than 20% of the water body. To overcome the spatial resolution differences among the diverse available products, all datasets are resampled to 0.25° spatial resolution by averaging the pixel values.

### 2.1.2 Reanalysis dataset and land surface model dataset

We collect the soil moisture data from ERA5, one global atmospheric reanalysis dataset released by the ECMWF (Balsamo

et al., 2015). The data assimilation system used for ERA5 is the ECMWF Integrated Forecast System (IFS), and the meteorological forcing for retrieving soil moisture is from the ERA atmospheric reanalysis. It can provide soils at four soil depths (0–7, 7–28, 28–100, and 100–289 cm). Here we select the daily averaged SM from the first soil layer to match with satellite CCI SM.

Daily potential evapotranspiration (PET) and surface soil moisture (0–15 cm) is collected from the Global Land-surface

Evaporation Amsterdam Methodology (GLEAM) dataset. GLEAM is based on a general land surface model that focuses on soil moisture and evapotranspiration (Miralles et al., 2011). PET in the GLEAM is calculated with the Priestley–Taylor formula based on multiple reanalysis datasets.

The soil moisture is calculated with a soil-water module based on water cycle balance.

Four meteorology variables are obtained from the Chinese regional ground meteorological dataset. They are precipitation, air

temperature, solar radiation, and wind. This dataset has a temporal resolution of 3-hourly and a spatial resolution of 0.1°. Considering the lag effect of precipitation on surface water dynamics, we use the five-day antecedent precipitation (AP) to replace the daily precipitation (Wei et al., 2020). Specifically, AP is estimated as follows:



$$\text{AP} = \frac{\sum_{i=1-m}^{i=0} P_i}{m}, \tag{1}$$

where $P_i$ is the precipitation at the day $i$; $m$ is the number of prior days for AP owing the highest absolute correlation
between SM and precipitation. Here we set $m$ as five based on the regional hydrological and climatic variability.

Three surface temperature sources are additionally collected for uncertainty analysis. Two sources are collected from the
ERA5 and GLDAS ensemble model. Considering the model uncertainties caused by regional surface characteristics and
climatic conditions, we simulate surface temperature and surface soil moisture (0–10 cm) by implementing one Noah model
that is forced with meteorology variables from the Chinese regional ground meteorological dataset and the surface condition
parameters from MODSI. This dataset is previously used in our work (Liu et al., 2020a; Liu et al., 2021b). All these surface
temperatures and soil moisture are resampled to a 0.25° grid.

### 2.1.3 In situ measurements

A variety of spatially sparse in situ soil moisture measurements is collected to evaluate the accuracy of gap-filled SM. We
collect in situ soil moisture observations at 39 sites obtained from the China Watershed Allied Telemetry Experimental
Research (WATER) project and the Chinese Ecosystem Research Network (CERN). These validation stations are set up in a
relatively large homogeneous area dominated by vegetation covers (cropland, woodland and grassland) or desert lands. In
addition, 657 in situ soil moisture measurements are collected from the Chinese agro-meteorological and ecological
observation network. These stations are covered by cropland. All these selected in situ soil moisture measurements have
been previously used for validating the satellite-derived soil moisture in China, and their locations and information are
displayed in Fig.1 and Table 2.

We also collect the dense in situ measurements at the Maqu soil moisture monitoring network. The Maqu network (33°30′–
34°15′N, 101°38′–102°45′E) is located on the north-eastern border of the Tibetan Plateau (Fig. 1) (Dente et al., 2012). In this
network, 20 sites are distributed over a uniform grassland cover, located in the large valley of the Yellow River. Maqu
network has demonstrated strong capability in monitoring the spatial and temporal SM variability with high accuracy. The
detailed information of available sites is summarized in Table 2.

**Table 2 Summary of the characteristics of in situ sites**

| ID | Site | Land-use | Elevation | Longitude | Latitude | Soil depth | Projections and references |
|----|------|----------|-----------|-----------|----------|------------|----------------------------|
| 1 | Yucheng | Cropland | 23m | 116.57E | 36.83N | 10cm | China Watershed Allied |
| 2 | Daxing | Cropland | 20m | 116.42E | 39.62N | 5cm | Telemetry Experimental |
| 3 | Miyun | Woodland | 350m | 117.32E | 40.63N | 5cm | Research (WATER), |
| 4 | Guantao | Cropland | 30m | 115.12E | 36.51N | 2cm | (Zhang et al., 2021a) |
| 5 | Arou | Grassland | 2995m | 100.46E | 38.04N | 10cm | (Li et al., 2009) |
| 6 | Maliantan | Grassland | 2817m | 100.30E | 38.55N | 5cm | (Huang et al., 2016) |

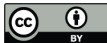



| | | | | | | | |
|---|---|---|---|---|---|---|---|
| 7 | Yingke | Cropland | 1519m | 100.42E | 38.85N | 5cm | |
| 8 | Guantan | Woodland | 2835m | 100.25E | 38.53N | 5cm | |
| 9 | AKA | cropland | 1008m | 80.85E | 40.67N | 10cm | Chinese Ecosystem |
| 10 | ALF | Woodland | 2455m | 101.02E | 24.54N | 5cm | Research Network |
| 11 | ASA | cropland | 1296m | 109.31E | 36.85N | 10cm | (CERN), |
| 12 | BJF | Woodland | 1162m | 115.43E | 39.97N | 5cm | (Yu et al., 2006) |
| 13 | BNF | Woodland | 722m | 101.02E | 21.95N | 10cm | (Li et al., 2018a) |
| 14 | CBF | Woodland | 512m | 127.09E | 42.40N | 5cm | (Zhu et al., 2007) |
| 15 | CLD | Desert | 1342m | 80.70E | 37.01N | 10cm | (Yao et al., 2018) |
| 16 | CSA | cropland | 21m | 120.38E | 35.25N | 10cm | |
| 17 | CWA | cropland | 1241m | 107.67E | 35.25N | 10cm | |
| 18 | DHF | Woodland | 412m | 112.53E | 23.17N | 15cm | |
| 19 | ESD | Desert | 1301m | 110.18E | 39.50N | 10cm | |
| 20 | FKD | Desert | 578m | 88.00E | 44.15N | 10cm | |
| 21 | FQA | cropland | 65m | 114.55E | 35.02N | 10cm | |
| 22 | GGF | Woodland | 6967m | 101.88E | 29.60N | 10cm | |
| 23 | HBG | Grassland | 3321m | 101.33E | 37.66N | 5cm | |
| 24 | HJA | cropland | 305m | 108.20E | 24.40N | 10cm | |
| 25 | HLA | cropland | 221m | 126.63E | 47.43N | 10cm | |
| 26 | HSF | Woodland | 102m | 112.90E | 22.70N | 10cm | |
| 27 | HTF | Woodland | 294m | 109.75E | 26.83N | 10cm | |
| 28 | LCA | cropland | 52m | 114.68E | 37.88N | 10cm | |
| 29 | LSA | cropland | 4230m | 91.33E | 29.66N | 5cm | |
| 30 | LZD | cropland | 1363m | 100.12E | 39.33N | 10cm | |
| 31 | MXF | Woodland | 2035m | 103.90E | 31.70N | 10cm | |
| 32 | NMD | Desert | 348m | 120.70E | 42.92N | 10cm | |
| 33 | QYA | cropland | 48m | 115.07E | 26.74N | 10cm | |
| 34 | SNF | Woodland | 1611m | 110.40E | 31.50N | 10cm | |
| 35 | SPD | cropland | 1413m | 104.95E | 37.45N | 10cm | |
| 36 | SYA | cropland | 35m | 123.40E | 41.52N | 10cm | |
| 37 | TYA | cropland | 62m | 111.50E | 28.91N | 10cm | |
| 38 | YGA | cropland | 448m | 105.45E | 31.27N | 10cm | |
| 39 | YTA | cropland | 44m | 116.92E | 28.25N | 10cm | |
| 40-59 | Maqu network | Grassland | ~3430m | 101.63-102.75E | 33.5-34.25N | 5cm | Tibetan Plateau observatory of plateau scale soil moisture |





| | | | | | | |
|---|---|---|---|---|---|---|
| | | | | | | and soil temperature (Tibet-Obs), (Su et al., 2013) (Wei et al., 2019) |
| 60-716 | Agro-meteoroloical stations | Cropland | -84-4200m | 75.98-134.28E | 18.5-51.72N | 10cm | China's agrometeorological observation network, (Meng et al., 2021) (Wang et al., 2016) |

## 3. Methods

Our study aims to reconstruct the CCI SM data gaps for obtaining spatially continuous records. The basic idea beneath the proposed gap-filling approach is to efficiently depict the correlation between the SM records and the corresponding
explanatory variables, which can be expressed as:

$$SM = f(V_1, V_2, V_3 \cdots\cdots V_k) + \varepsilon, \tag{2}$$

$$V_i \in R^{N,T}, \tag{3}$$

where $SM$ is the soil moisture, $V_i$ is the corresponding explanatory vectors, and $k$ is the number of the input variables. $V_i$ can be a vector the sample number of which is decided by the spatial domain ($N$) and temporal domain ($T$). f is one function that
can be either linear or nonlinear. $\varepsilon$ represents the model residual. In machine learning ensemble, $f$ represents a black box model that does not have one specific form.

Proposed methodology mainly involves the following steps: (i) using a regression subset selection model and a variable correction procedure to filter explanatory variables from the satellite observations and model-driven knowledge, and removing the systematic bias between them; (ii) applying a random forest algorithm to delineate the SM-explanatory
variables correlation based on the available pixels identified with a spatiotemporal window search strategy, and then employing the established correlation to retrieve the unavailable SM pixels; and (iii) conducting a geographically weighted regression and Gaussian filtering to calibrate the model-derived residuals. Figure 2 shows the overall diagram of our work.





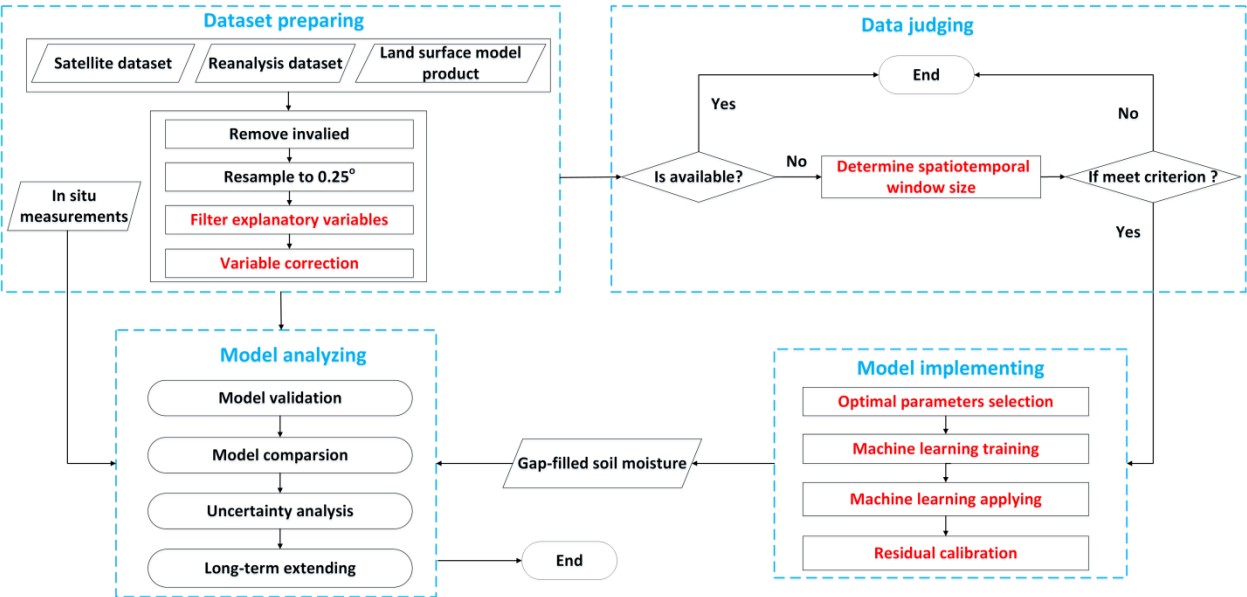

**Figure 2: The overall diagram of the proposed methodology. The main work carried out includes dataset preparing, data judging,**
**model implementing, and model analyzing.**

## 3.1 Explanatory variables filtering

### 3.1.1 Explanatory variables

Explanatory variables related to atmospheric, geophysical, ecological and hydrological variables are conductive in capturing
SM variability. The main assumption beneath this approach is that the suppressor variables are associated significantly with
each other in regression models, although they may be less correlated with the dependent variables. The significance
percentage produced in the regression subset selection model (Fu et al., 2019; Liu et al., 2021a) is employed to measure the
impacting probability of explanatory variables, and the high significance percentage indicates the strong ability to depict SM.
To be specific, we successively (1) use a least-squares linear regression to check the potential relationships between SM and
explanatory variables; (2) apply a stepwise regression to explore plenty of potential explanatory variables based on the
Akaike Information Criterion (AIC); (3) exploit the best models from all variable combination to identify the important
variables impacting SM; and (4) quantify the relative contributions of each explanatory variable to SM based on importance
criterion.

We conduct the subset selection model analysis based on the dataset during 2005—2015. 15 variables are selected as input
parameters, including seven surface environmental variables, i.e., Albedo, NDVI, EVI, LAI, DTR, PET and ERA SM, three
elevation variables, i.e., TPI, aspect and slope, three climatic variables, i.e., AP, air temperature, wind, and two geographical
factors, i.e., latitude and longitude. Notice that all these variables can be available from a reliable dataset at the continental



scale. Moreover, these variables have been reported previously (Cui et al., 2016; Liu et al., 2020b; Almendra-Martín et al., 2021) in robustly describing soil moisture.

As illustrated in Fig. 3(a), Albedo, NDVI, EVI, LAI, DTR, AP, PET, ERA SM, TPI and air temperature have the highest
significant percentage in correlating to CCI SM. Considering EVI and LAI have closely correlated to NDVI, and air temperature has closely correlated to DTR. EVI, LAI and air temperature are excluded for model application. All these selected covariates are physically meaningful in depicting SM. Atmospheric variables (i.e., precipitation, DTR and PET) are suitable to capture the temporal dynamics of SM. Topographic variables are included to both depict the orographic effects and recapture the spatial pattern of SM. NDVI and Albedo (and DTR) are included owing to their impact on the formation of
SM. Specifically, DTR exhibits a substantial correlation with SM owing to its capacity in taking account for land-atmosphere coupling. ERA surface moisture is also included to reproduce satellite SM. Despite uncertainties in the numerical model SM simulations, ERA SM provides the fewest dataset gaps. We further investigate the contribution of these variables to SM. As shown in Fig. 3(b) (and Fig. S1), the correlations between the CCI SM and the selected variables are relatively high across the whole of China. This indicates the feasibility of the selected variables in modeling SM. In addition,
since these variables are derived from optical remote sensing, reanalysis dataset and land surface model products, they have the potential to extend to large regions due to high availability (Fig. 3(c)).

### 3.1.2 Variable correction

To make the modeled values (i.e., ERA SM) comparable to satellite observations (i.e., ESA CCI SM), it is necessary to remove the systematic bias between them. Here we use one correction procedure (Long et al., 2020; Zhang et al., 2021d),
which primarily combines the variance scaling algorithm and the linear scaling algorithm. The used procedure can be illustrated with the following equations:

$$\begin{cases} SM_{c1} = SM_{ERA}(t_{av}) - \mu\big(SM_{ERA}(t_{av})\big) + \mu\big(SM_{ESA}(t_{av})\big) \\ SM_c = \mu(SM_{c1}) + (SM_{c1} - \mu(SM_{c1})).\sigma\big(SM_{ESA}(t_{av})\big)/\sigma\left(SM_{c1} - \mu\big(SM_{p1}\big)\right) \end{cases}' \tag{4}$$

where $SM_{ERA}$ is the raw ERA SM time series of the target grid pixel; $t_{av}$ is time series in which pixels in the object grid are available; $SM_{ESA}$ is the SM of the grid; $\mu$ and $\sigma$ are the mean value and the standard deviation, respectively. $SM_c$ is the
corrected ERA SM that is assumed to have a spatial pattern (i.e., consistent means and standard deviations) with the CCI SM. In our study, a dataset regarding the time series of 2005—2015 is used to conduct the correction procedure for guaranteeing enough samples. The examples illustrating the performance of ERA SM correction can be found in Fig. S2. Despite being conducted on SM parameters here, this calibration method can be implemented to other parameters (e.g., DTR) when replaced with numerical model outputs.



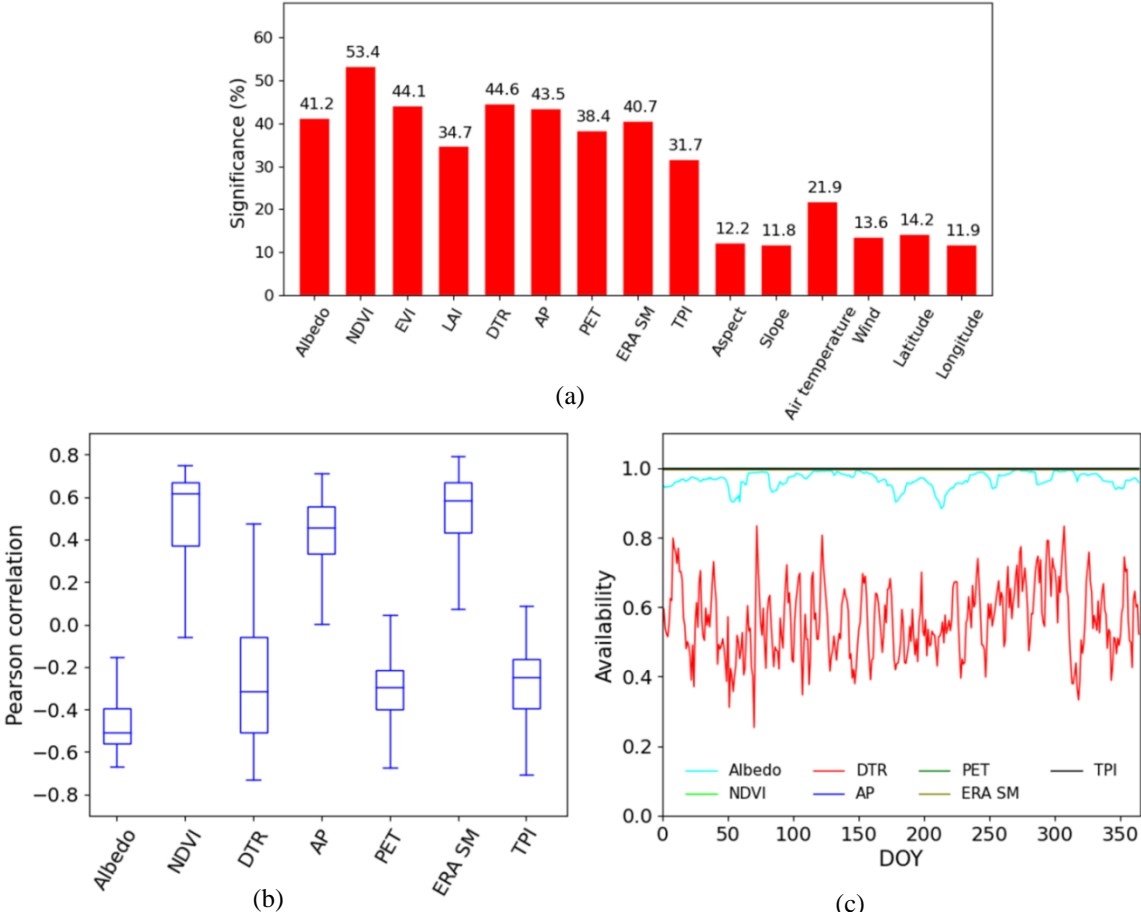

(a)

(b)

(c)

**Figure 3: The correlation and availability of dataset used. (a) The significance percentage of the selected variables in correlation to CCI SM. (b) The Pearson correlation between the selected variables and CCI SM. (c) The availability of the selected variables.**

### 3.2 Random Forest regression

Despite being easy to implement and requiring less computational resources, the traditional regression-based methods cannot provide full probability density functions for evaluating system performance. Ensemble learning approaches may overcome this issue by aggregating results from multiple models. Machine learning approaches could achieve better performance owing to their capacity in reducing overfitting chances and quantifying the accompanying uncertainty. Among the machine learning models, random forest is one effective and powerful tool in interpreting earth variables, acting as an enhanced decision tree model (Belgiu and Drăguţ, 2016). As illustrated in Fig. 4, RF is a hierarchical tree diagram, which is based on a nonparametric strategy and therefore is feasible to add layer categories. This decision tree model is composed of many nodes and edges within each tree structure, mainly including two types of nodes: split nodes and leaf nodes. The split node is related to a test function that is employed to split the input data, whereas the leaf node is associated with the final decision. Unlike the standard decision tree model that relied on the whole data set, RF trains each tree on bootstrap resamples. This





model only considers the randomly selected variables rather than the total variables. By this means, the outcome is decided

by one majority voting or averaging strategy.

In this study, the RF model is implemented using the function 'RF Regressor' from the Python Library (Shahriari et al., 2016). Specifically, the built-in functions are used to assess the importance of each covariate by using the out-of-bag samples. We use the 'Bayesian Optimization' module (http://rmcantin.github.io/bayesopt/html/bopttheory.html) to select the best hyperparameters in driving RF algorithm. Four critical parameters deciding the RF algorithm include the number of

trees (n_estimators), the maximum tree depth (max_depth), the minimum number of samples for splitting an internal node (min_samples_split), and the number of features (max_features). For each specific climate region, the Bayesian optimization process is carried out within 20 iterations to optimal parameters. The training procedure is mainly based on the dataset covering 2003—2008. Optimal parameters in the seven climate regions are listed in Table 3.

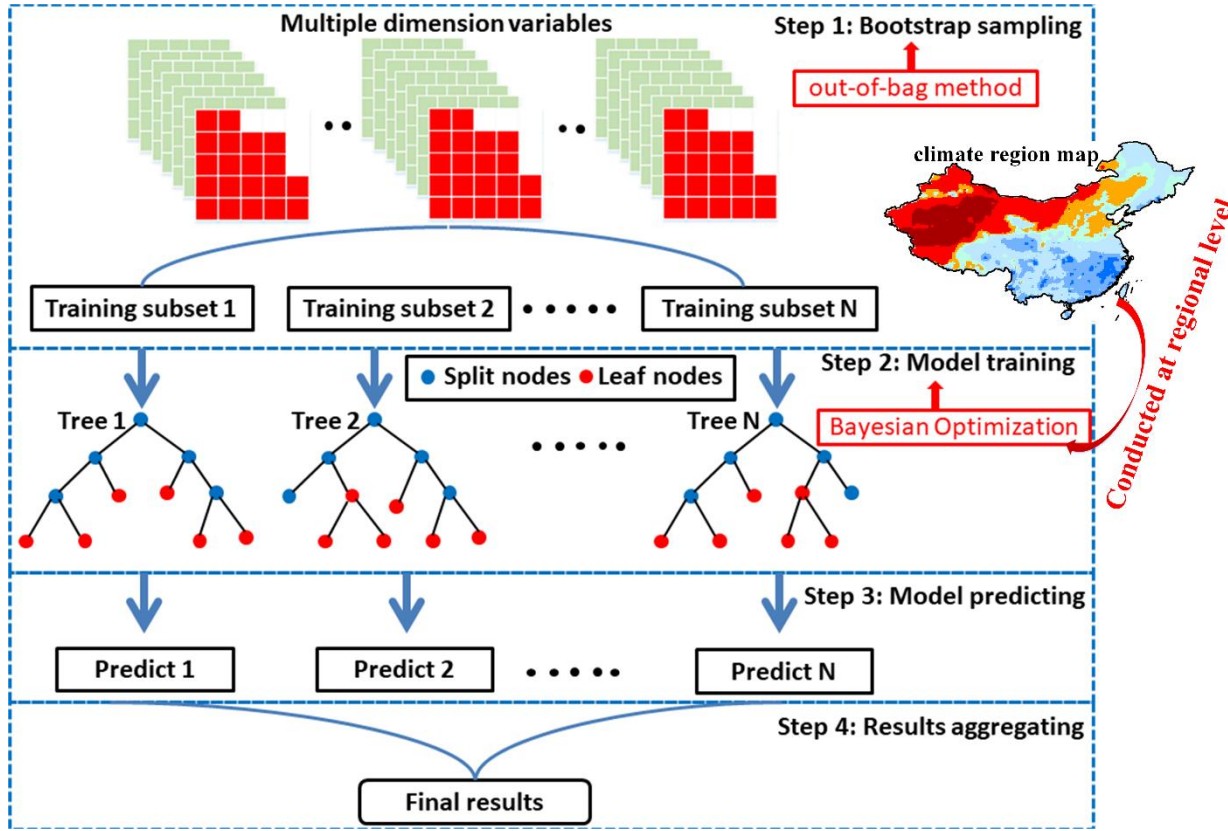

**Figure 4: The diagram of the random forest model implemented for a multidimensional dataset. Stage 1: The out-of-bag sampling method is used to sample time series dataset. Stage 2: The independent decision trees for model training are built based on the optimal parameters from the Bayesian Optimization module. This process is conducted at the climate regional level. Stage 3: The predicted values are produced from each bootstrap tree. Stage 4: The outcome is obtained by averaging earlier results from multiple trees.**






**Table 3 Optimal parameters regarding seven climate regions**

| Climate region | n_estimators | max_depth | min_samples_split | max_features |
| --- | --- | --- | --- | --- |
| Arid | 69 | 11 | 8 | 0.12 |
| Semi-arid | 80 | 18 | 9 | 0.16 |
| Aird/semi-wet | 47 | 9 | 5 | 0.31 |
| Wet/semi-arid | 36 | 10 | 3 | 0.25 |
| Wet | 52 | 15 | 11 | 0.16 |
| Moist | 62 | 10 | 9 | 0.12 |
| Over-wet | 22 | 8 | 4 | 0.27 |

### 3.3 Spatiotemporal strategy

One critical issue of the machine learning model is how to efficiently explore the informative covariates. Here we use one
spatiotemporal strategy to most capture the spatial and temporal SM and the related covariate dynamics. Our strategy
primarily relies on the available pixels within one regional subset, thus including more interest pixels to participating
regression. Figure 5(a) provides the diagram of the spatiotemporal window search strategy.

One adaptive strategy is employed to determine the optimal spatiotemporal window size. Two critical variables are adopted
to identify the window size, i.e., the size of the spatial window (sw) and the number of temporal days (nd). To find the
optimal sw and nd, we continually increase the value of sw and nd from the initial values until the samples participating for
regression meet the criterion, i.e., the number of available pixels within the searched window should be no less than eight
times of the participating explanatory variables (i.e., seven) (Svetnik et al., 2003; Liu et al., 2020a). Here an initial sw is set
to 5 and an initial nd is set to 1. Considering that a fraction of gaps occur in the satellite dataset (e.g., LST and albedo) and
the optimal window may not exist, the maximum values of sw and nd are introduced to terminate this process. One
sensitivity analysis is conducted with the independent dataset to select the two maximum values. Specifically, we conduct
one cross validation during 2003—2008 to evaluate the accuracy of the gap-filling model. The increasing maximum nd from
1 to 7 with intervals of length 1 is tested, and the maximum sw is tested from 4 to 10 with intervals of length 1. The values
that yield the lowest RMSE (Fig. 5(b)) are selected, and finally, we set maximum sw to 7 and the maximum nd to 4. Note
that we also conduct sensitivity analysis for each climate region and find no substantial differences in the resulting optimal
values of two parameters among seven climate regions. This is probably because this sensitivity analysis is more reliant on
model structure rather than sample characteristics.



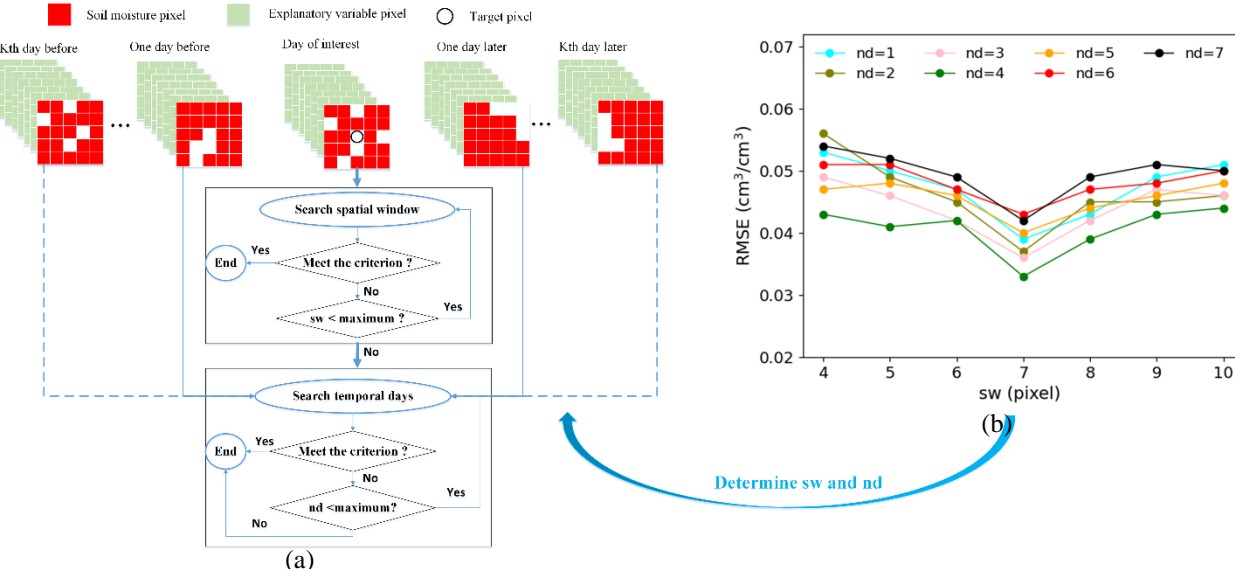

**Figure 5: (a) The diagram of spatiotemporal window determination strategy for random forest regression. (b) The results of the sensitive analysis regarding two maximum values for terminating the searching process.**

**3.4 Residuals calibration**

Considering the machine learning model may not fully account for the variability in soil moisture, the original reconstruction needs to be calibrated with the corresponding model residuals. The calibration procedure can correct the bias resulting from neglectful variables (Zhu et al., 2012; Liu et al., 2020a). This is very critical considering the relatively less spatial representative of soil moisture within the 25 km scale. In practice, we add the interpolated residuals to the original

reconstructions.

A GWR model (Li et al., 2017) is applied to interpolate the RF-derived residuals. This procedure is based on the samples within the searched window for each target pixel. The model residual ($\varepsilon_j$) derived from Eq. (2) can be described using the explanatory variables as follows:

$$\varepsilon_j = \beta_0(u_j, v_j) + \sum_{i=0}^{k} \beta_i(u_j, v_j) X_{ij}, \tag{5}$$

where $\beta_0(u_j, v_j)$ and $\beta_i(u_j, v_j)$ are the regression coefficients estimated at the $j$th pixel, and $(u_j, v_j)$ are the coordinates. The regression coefficients can be estimated using the observations within the self-adaptive searched window as follows:

$$\begin{cases} \hat{\beta}(u_j, v_j) = (X^T(W(u_j, v_j))X)^{-1} X^T W(u_j, v_j) Y \\ \qquad\qquad w_{ij} = [1 - (d_{ij}/b)^2]^2 \end{cases}, \tag{6}$$

where $\hat{\beta}(u_j, v_j)$ is the coefficient matrix composed of coefficients from each explanatory variable; $X$ and $Y$ are the explanatory variable matrix and the dependent variable (i.e., SM) vector, respectively. Her latitude, longitude and seven





explanatory variables selected in section 3.1.1 are used to implement the GWR model. $W(u_j, v_j)$ is the weight matrix

composed of $w_{ij}$, $d_{ij}$ is the Euclidean distance between the observation $ith$ and the $j$th point, a and $b$ is the window radius.

Before adding to the original reconstruction, the GWR interpolated residual is further smoothed with a normalized k × k

Gaussian filter with a standard deviation of σ. This procedure can remove the grid-like artifacts that extensively exist in

statistical model outcomes. Base on the optimization procedure (Sismanidis et al., 2021; Liu et al., 2019), we set k = 5 and σ

=1.5.

### 3.5 Accuracy evaluation

The top layer (Table 2) SM measurements from in situ stations are used to evaluate the accuracy of the reconstructed results.

Considering the scale mismatch between the sparse in situ station and CCI SM product (~25 km), we use the Disaggregation

based on Physical And Theoretical scale Change (DISPATCH) model (Merlin et al., 2012) to disaggregate the 0.25°

reconstructions to 1 km resolution. As one typical SM disaggregation model, DISPATCH has been extensively applied in

current studies (Molero et al., 2016; Song et al., 2021). In DISPATCH, SM can be depicted at different spatial scales by

linking to LST and NDVI through soil evaporative efficiency. Detailed descriptions regarding this disaggregation method

can be found in Supplementary Text S1.

In addition to field measurements, one holdout cross-validation with ten replicates is conducted to comprehensively evaluate

the model performance. For each replicate, we separate the dataset (CCI SM and explanatory variables) into the training

(90%) and the test subsets (10%). After the gap-filled SM series are produced with the training set, they will be validated

with the test set. The above cross-validation is also applied to conduct model comparison and uncertainty analysis.

The statistics used for the model accuracy assessment include the coefficient of determination ($R^2$), the root mean square

error (RMSE), the mean absolute error (MAE), the average error bias (BIAS), and the unbiased RMSE (ubRMSE). All these

metrics have been extensively used for evaluating satellite SM, and they can be described as follows:

$$R^2 = 1 - \frac{\sum_i^k (SM_i - \widehat{SM_i})}{\sum_i^k (SM_i - \overline{SM})}, \tag{7}$$

$$RMSE = \sqrt{\frac{\sum_i^k (SM_i - \widehat{SM_i})^2}{k}}, \tag{8}$$

$$MAE = \frac{\sum_i^k |SM_i - \widehat{SM_i}|}{k}, \tag{9}$$

$$BIAS = \frac{\sum_i^k (\widehat{SM_i} - SM_i)}{k}, \tag{10}$$

$$ubRMSE = \sqrt{RMSE^2 - BIAS^2}, \tag{11}$$

where $SM_i$ is the reconstructed soil moisture of the $i$th sample, and $\widehat{SM_i}$ is the corresponding reference (or field) value, $k$ is

the number of samples.





## 4. Results and discussions

### 4.1 Spatiotemporal patterns

The spatiotemporal pattern of the original daily CCI SM and the corresponding gap-filled dataset in 2009 is first checked. As shown in Fig. 6(a) (and (Fig. S3)), a considerably large gap occurs in the original CCI SM, and this gap issue is heavier in the winter season. We reconstruct the contaminated SM pixels using the spatiotemporal random forest model. It's observed that most of the contaminated pixels (more than 85%) are reconstructed. Relatively fewer missing pixels are gap-filled in the winter season in comparison to other seasons, primarily relating to the heavy missing issues during this time.

Figure 6(b) shows the boxplot of original versus gap-filled SM on the selected days in 2009. Relative minor conformity exists between the original and reconstructed SM for most days. A consistent pattern between them is also observed on the monthly-average SM, as illustrated in Fig. 6(c). Large differences occur in the winter and spring seasons. This can be attributed to the fact that the original CCI SM provides less training data from October to May of the following year. In addition, the distribution of CCI SM is more uneven in this period, which may reduce the model performance due to the

limited representation of training samples (Stroud et al., 2001).

We further check the model performance regarding different climate regions. Figure 6(d) demonstrates one minor discrepancy between the original and the reconstructed SM, with the bias in median SM values less than 8%. This means the reconstructed SM owns a strong variation delineating capacity. A small overestimation occurs for the arid regions, which originally have less soil water storage.



**Figure 6: The comparison between CCI dataset and gap-filled SM in 2009. (a) The plots of the availability of CCI dataset and gap-filled SM. (b) The boxplot of the CCI dataset and gap-filled SM on the selected days. (c) The boxplot of month-average CCI and gap-filled SM. (d) The boxplot of raw and gap-filled SM regarding seven climate regions.**

Figure 7 exhibits the spatial distributions of the original CCI SM on the selected days in 2009. It's observed that the humid regions are mostly concentrated in southern China that is adjacent to the west coast of the Pacific, whereas the dry regions are mainly distributed in the northern and western parts. A considerable fraction of contaminated pixels is observed on the selected days, and this contamination issue is severe in the winter season and mountainous areas (e.g., Tibet Plateau and Mongolian Plateau). The spatial distributions of the reconstructed SM corresponding to the selected days are illustrated in Fig. 8. Almost all the contaminated pixels from March to October are reconstructed; meanwhile, the proposed model reconstructs the most contaminated pixels for the remaining months. Owing to the additional valid values provided by gap-filled pixels, more spatial variations are delineated in the reconstructed SM images. The missing pixels still occur in the reconstructed SM images especially in the cold seasons. Some of these invalidate pixels correspond to the snow and water-covered regions that have been beforehand removed.







**Figure 7: The spatial distributions of raw CCI SM on the 15th of each month in 2009.**



**Figure 8: The spatial distributions of gap-filled CCI SM on the 15th of each month in 2009.**

## 4.2 Accuracy validation

The proposed model is first evaluated with sparse in situ measurements from WATER and CERN. To avert the mismatch
issue, the 25 km SM dataset is disaggregated to 1 km using the DISPATCH model before accuracy validation. As shown in
Fig. 9(a), good accordance is obtained between 1 km CCI SM-derived values and in situ measurements, with an $R^2$ of 0.8.
This accordance is also found between the 1 km reconstructed SM and in situ measurements (Fig. 9(b)), with the $R^2$ of 0.75.





Both the CCI SM-derived values and reconstructed SMs are close to the 1:1 line when evaluating with in situ measurements. High accuracies are observed when evaluating with in situ measurements from the national agro-meteorological stations.

Fig. 9(c) and (d) show the $R^2$ between 1 km CCI SM-derived values and in situ measurements is 0.81, while the $R^2$ between 1 km reconstructed SM and in situ measurements is 0.71. In general, our model is capacity in reconstructing SM. The inconsistency still exists, and noticeable overestimations are observed in high SMs.

We further validate the reconstructed results with the dense in situ measurements from the Maqu network. The RMSE and MAE is 0.11 and 0.09 cm$^3$/cm$^3$ (Fig. 9(e)) for the 1 km CCI SM-derived values, respectively, and is 0.12 and 0.09 cm$^3$/cm$^3$

(Fig. 9(f)) for the 1 km reconstructed SM, respectively. This means a good agreement is obtained for both the CCI SM product and gap-filled SM. The poor performance is found in the low values, mostly due to the extreme conditions and the fewer samples for model regression.

The time series of average 0.25º CCI SM values and reconstructed SM over the dense grid are compared to the dense in situ observations. Both the original and reconstructed SM matches well with the in situ series, with the NSE of 0.83 and 0.85,

respectively. The reconstructed SM (Fig. 9(g)) mostly describes the temporal dynamics of in situ measurements, i.e., sufficiently capturing seasonal and daily variability. It is also observed that the rainfall events impacting the surface dynamics are well delineated on the SM temporal variations. In general, the reconstructed SM seems to have inherited the merits of stability between April and November from CCI SM, i.e., having comparable values during this period. This is reasonable, since in addition to focusing on common explanatory variables such as Albedo, NDVI and DEM, our method

introduces time series water-heat components, i.e., precipitation, PET, and reanalysis soil moisture, providing a substantial contribution in reconstructing time series SM. Meanwhile, since the used variables are daily, our model shows strong capacity in delineating abrupt climatic changes (Piles et al., 2016).



**Figure 9: The evaluations of model results. (a), (c) and (e) are the scatter plots of 1-km CCI SM-derived values against field**
**measures regarding WATER/CEERN, agro-meteorological stations, and Mauqu network, respectively, and (b), (d) and (f) are the**
**scatter plots of 1-km gap-filled SM-derived values against field measures. (g) are the time series of average CCI SM-derived values**
**against site measures in the Maqu region. The shaded area in (g) denotes ±1 standard error.**

One cross-validation analysis is further conducted to evaluate the model performance. The obtained metrics (Fig. 10(a))

illustrate a good coincidence between the reconstructed and original CCI SM, with the median $R^2$ range between 0.51 and

0.63. Better accuracies are also demonstrated in the metrics of RMSE, MAE and ubRMSE. In particular, the medians of

BIAS are less than 0.01 cm$^3$/cm$^3$. Relatively better accuracies happen in the growth seasons (March-October). This can be

attributed to the fact that the critical environmental factors, such as NDVI, DTR and ERA soil moisture, are more related to

soil moisture during the vegetation growing seasons (Chen et al., 2014; Otkin et al., 2016).





Figure 10(b) shows the accuracy metrics for different climate regions. A similar pattern as those in monthly means is
observed, i.e., acceptable accuracy occurs in most regions. No significant differences in median $R^2$ and BIAS happen
between the reconstructed SM of each climate region, with the bias between the maximum and minimum median $R^2$ and
BIAS being less than 0.09 and 0.003 cm³/cm³, respectively. The relatively poor performance of metrics (e.g., RMSE and
MAE) in wet regions can be related to the high values in albedo and specific heat capacity (Guan et al., 2009). Meanwhile,
the fewer amounts of available observations (i.e., LST and albedo) in these areas can affect the model capacity and stability.
It should be noticed that, despite the relatively high RMSE, MAE and ubRMSE over the humid region, $R^2$ is very high, as
illustrated in Fig. 10. This might be due to the SM variability in these areas being high.

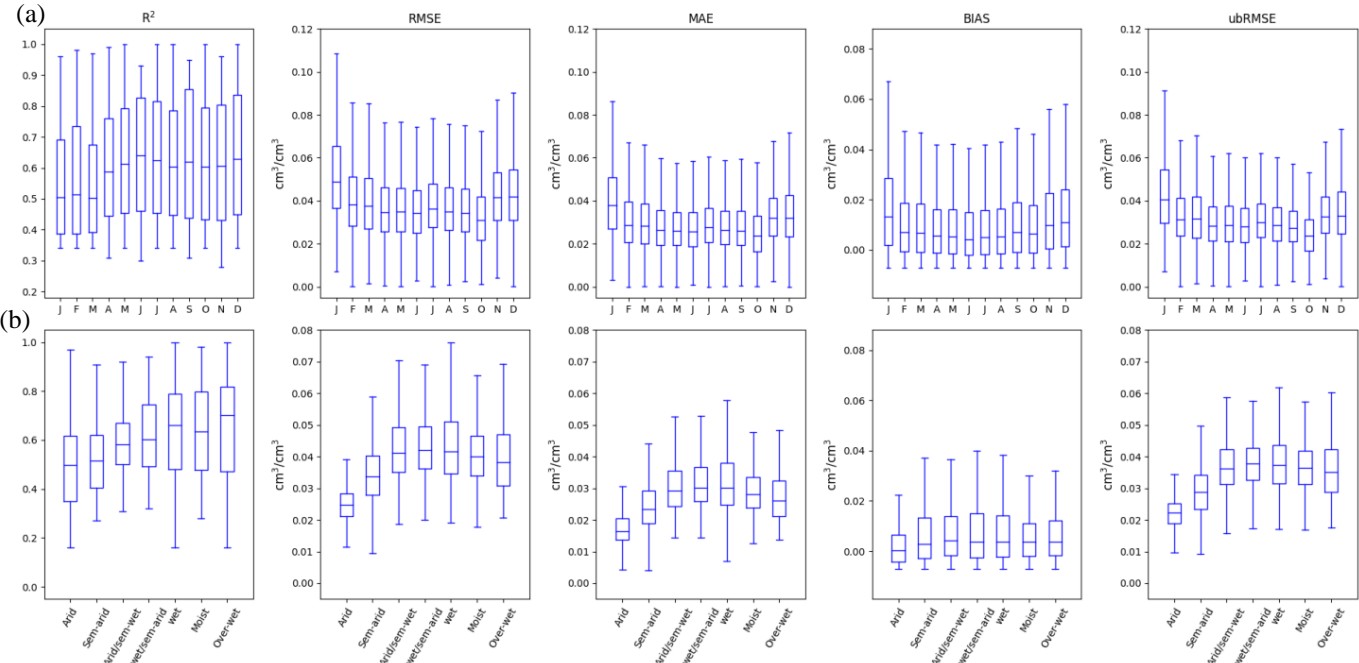

**Figure 10: The accuracy metrics of 10-cross validation for R2, RMSE, MAE, BIAS, and ubRMSE: (a) is averagely obtained on a
month basis, and (b) is averagely obtained for each climate region.**

The spatial distributions of accuracy metrics in Fig. 11 further illustrate the good accuracy of the proposed gap-filling model.
The obtained metrics in the cross-validation analysis show a good match between the reconstructed SM and the original SM
CCI for most of the grids. Discrepancies are observed in some grids, but they rarely exceed 0.09 cm³/cm³ in absolute value.
Spatially, the distribution of reconstructed SM follows a geographic gradient. The relatively lower accuracies occur on the
complicated terrain in western China. For these regions, complex atmospheric conditions caused by high elevations tend to
affect the delineations of surface parameters. Meanwhile, complex topography may result in a complicated directional
anisotropy, bringing more uncertainty in modeling surface energy and water cycles (Hu et al., 2016).

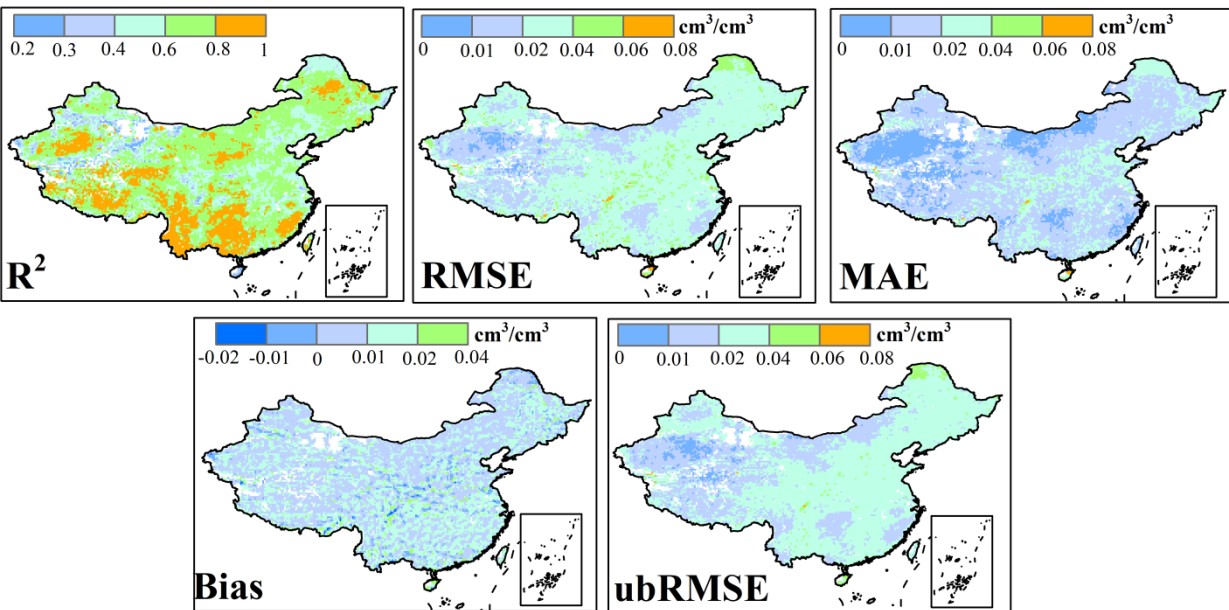

**Figure 11: The spatial distributions of accuracy metrics of 10-cross validation in 2009 for R2, RMSE, MAE, BIAS, and ubRMSE.**

## 4.3 Comparison analysis

To fully reveal the merits in modeling the critical surface characteristics, our model is further compared against previous studies, as illustrated in Table 4. In general, the accuracies produced by our model are comparable and even better than previous studies, despite being evaluated with different in suit measurements and simulated approaches. The satisfied performance of the proposed model is plausible, which can be attributed to the following aspects. Given the complex underlying surface and the diverse climatic conditions across one continuous scale, we chose more efficient explanatory
variables relative to previous research. Especially, the land surface model and reanalysis outputs are introduced to bring additional context for overcoming the severe missing issues of remotely sensed variables. Simultaneously, an adaptive spatiotemporal domain strategy and residual calibration module are incorporated into machine learning to balance the regression performance, overfitting problems, and computational complexity. This strategy focuses on the covariates that include both the spatial and temporal domains, therefore possessing the potential of producing more reasonable accuracy
compared to other approaches that utilize either spatial domain (Li et al., 2021a).

**Table 4 The accuracy comparison among different literatures**

|  | SM source | Model | Adopted variables | Study Region | Accuracy | | Literatures |
|---|---|---|---|---|---|---|---|
|  |  |  |  |  | Field-validation | simulation-validation |  |
| 1 | ESA CCI | RF | LST, Precipitation, NDVI, PET, Soil texture | Southern Europe |  | RMSE=0.024-0.025 m³/m³ | Almendra-Martín et al. (2021) |



| | | | | | | | |
|---|---|---|---|---|---|---|---|
| 2 | ESA CCI | ANN | Precipitation, Temperature, NDVI, LAI, DEM, Slope, Aspect, Latitude, Longitude, Soil texture | China | RMSE=0.036-0.074 m³/m³ | RMSE=0.037-0.064 m³/m³ | Zhang et al. (2021b) |
| 3 | ESA CCI | RF | NDVI, LST, Daytime LST, Nighttime LST, Diurnal LST | Oklahoma, USA | RMSE=0.08 m³/m³ | RMSE=0.02 m³/m³ | Liu et al. (2020b) |
| 4 | AMSR2 | CNN | / | Global | RMSE=0.097 m³/m³ | RMSE=0.065-0.073 m³/m³ | Zhang et al. (2021c) |
| 5 | ESA CCI | MLR, OK, RK | Precipitation, Temperature | Midwest, USA | RMSE=0.067-0.070 m³/m³ | | Llamas et al. (2020) |
| 6 | FY-3B | BP-NN | LST, NDVI, Albedo, Latitude, Longitude, DEM, DOY | Tibetan Plateau | RMSE=0.1 cm³/cm³ | | Cui et al. (2016) |
| 7 | ESA CCI | RF | Albedo, NDVI, DTR, AP, PET, ERA SM and TPI | China | RMSE=0.09-0.14 cm³/cm³ | RMSE=0.025 cm³/cm³ | Our study |

Note: CNN = Convolutional neural network, OK = ordinary kriging, RK = regression kriging, BP-NN = back-propagation neural network

The proposed method is future compared against four extensively used models that adopt the same explanatory variables and spatiotemporal window search strategy. The first one is the conventional multiple linear regression (MLR) approach. Three typical machine learning approaches, i.e., Extreme gradient boost (XGB), Support vector machine (SVM) and Artificial Neural Network (ANN), are also used for comparison. Detailed descriptions of four available models can be found in supplementary Text S2. The accuracy metrics of the five models are shown in Fig. 12. Generally, the MLR, XGB, SVM and ANN, accompanying RF, could potentially reconstruct the missing CCI SM pixels, indicating the stable suitability of these models and the feasibility of available variables. Moreover, the RF model demonstrates prominent performance among all the models, further manifesting its capacity in reconstructing soil moisture when integrating the effective dataset source and mining method. This is consistent with earlier studies that illustrate the robustness of RF in simulating satellite parameters (Karbalaye Ghorbanpour et al., 2021; Zhao et al., 2018). Based on this, we also check the accuracy of models excluding the residual calibration procedure, which is one essential procedure for the proposed model. Results in Fig. 12 demonstrate the accuracies are lowered by ~9% when removing residual calibration, underscoring the importance of residual modulation in improving SM reconstructing. Meanwhile, the better performance brought by spatiotemporal domain strategy is also exhibited when compared with the global regression, as illustrated in Fig. 12. Quantitatively, the spatiotemporal domains can improve the accuracy of ~19% in forcing RF regression. Overall, these analyses indicate the feasibility of the proposed model by integrating the modules of residual calibration and spatiotemporal domain strategy.





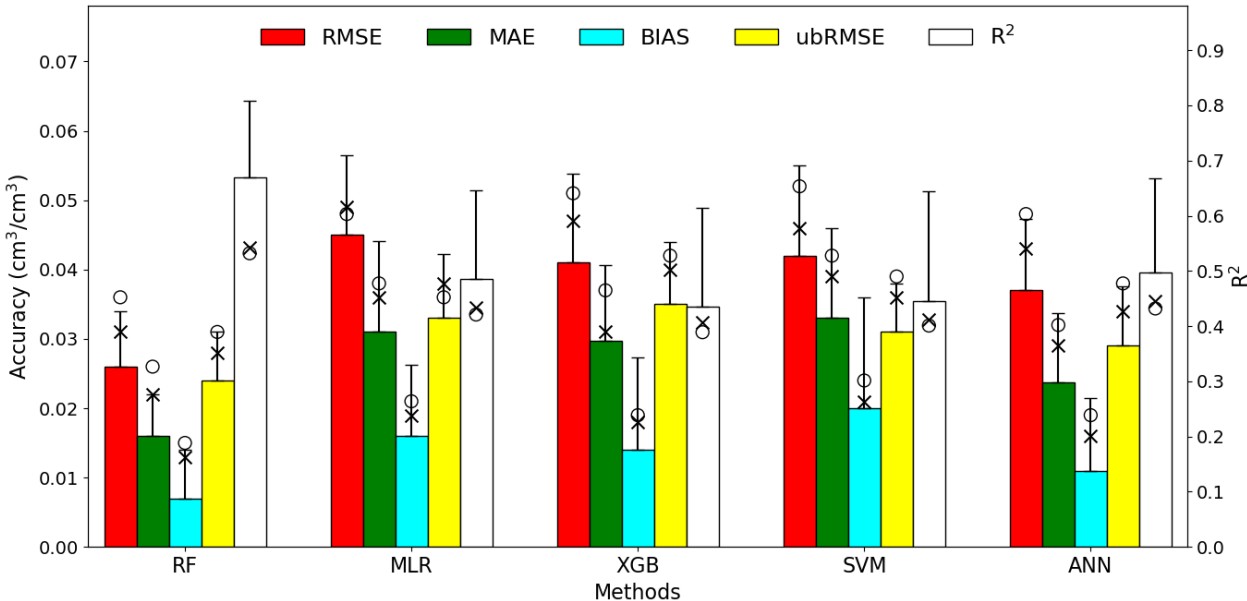

**Figure 12: Comparison RF-based model with other models (i.e., MLR, XGB, SVM and ANN). Error bars denote 1σ errors. The symbol 'x' represents the accuracy metrics of models excluding the residual calibration procedure, and the symbol 'o' represents the accuracy metrics of the models that use the global regression rather than regional regression based on the spatiotemporal window searching strategy.**

## 4.4 Importance analysis

Considering the criticality of explanatory variables in simulating SM, importance analyses regarding these selected variables are conducted. We first investigate the accuracy of the reconstruction model that excludes one participating variable. As illustrated in Fig. 13(a), the performance of the model with six variables (i.e., excluding one) is relatively lower when compared with that with seven variables. The removing-one strategy can lower the accuracies of 2.2-6.4% in $R^2$ and 10-30% in BIAS. This diminished performance is plausible since SM is heavily related to all the selected variables. Specifically, variability in land surface characteristics (NDVI and albedo) and atmospheric conditions (i.e., precipitation and PET) can heavily impact the soil moisture variability. Meanwhile, additional covariates mean an increase in the number of samples participating in the regression model, therefore potentially resulting in an improvement of overall accuracy. Specifically, we observe the lowest accuracy occurs when DTR is excluded, underscoring the vital role of DTR in modeling SM.

We further investigate the contribution of each explanatory variable in modeling SM. The importance score produced by the RF algorithm itself is used to delineate relative contribution (Zhao et al., 2019b; Ramoelo et al., 2015). As demonstrated in Fig. 13(b) (and Fig. S4), AP and ERA SM substantially impact CCI SM modeling, with the average importance score of 0.48 and 0.47, respectively. The NDVI, albedo, TPI and PET have less importance with the average score of 0.41, 0.43, 0.42 and 0.45, respectively. Specifically, the DTR shows the highest importance with an average score of 0.52. This is mainly related to the fact that temperature variations likely have a far-reaching influence on soil moisture fluctuation. The heat from the





surface can be transferred to the atmosphere via evapotranspiration and sensible heat conduction, thereby modifying surface

soil moisture variations (Amani et al., 2017).

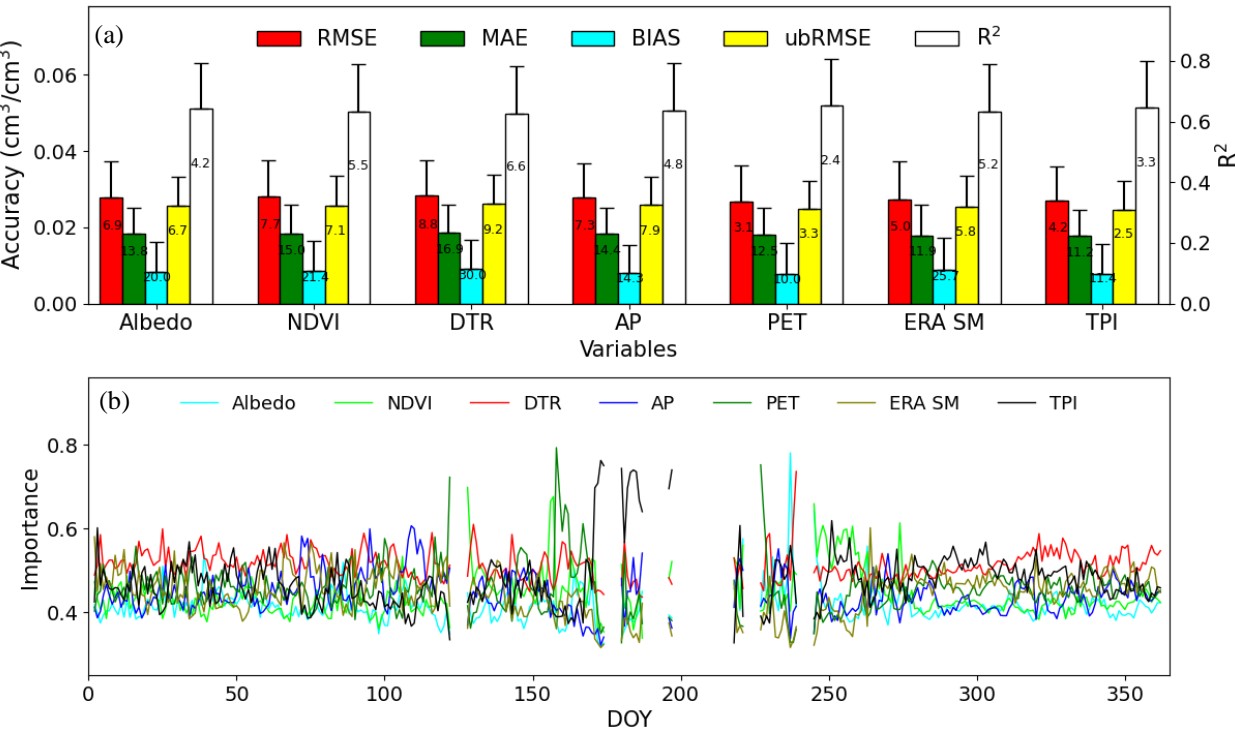

**Figure 13: (a) The accuracy of the models removing one variable, i.e., using other six variables in model regression. Error bars denote 1σ errors. The text denotes the relative percentage of the lowered accuracy in comparison to those using all seven variables.**
**(b) The importance score of the selected variables derived from RF regression.**

**4.5 Uncertainty analysis**

Given the critical importance of satellite-derived DTR and the severe missing issues in satellite-observed LST products, we

further investigate the substitution performance of other surface temperature sources in reconstructing SM, i.e., i.e., Noah,

ERA and GLDAS. This analysis is conducted at two focused regions (in Fig. 1) that have sufficient data sources supporting

our experiments: one region is in northern China that is mostly occupied by arid and semi-arid areas, while the other region

is in southern China that is occupied by wet areas. Considering the bias between satellite LST and modeled surface

temperature, the variable correction described in section 3.1.2 is conducted to remove the systematic bias and make the

simulated DTR comparable to satellite observations.

Figure 14(a) shows the minor reductions in Pearson correlation and RF-derived importance score of three numerical model

simulated DTRs when comparing with MODIS-derived DTR, which indicates the feasibility of all these datasets in

reconstructing SM. Fig. 14(b) and (c) illustrate noticeable reductions in model accuracy when replacing the satellite LST

with the other three simulated sources. Regarding the accuracy metric, the RMSE of reconstructed SM can be reached to

0.0282, 0.0299 and 0.0303 cm$^3$/cm$^3$, when respectively using Noah, ERA and GLDAS-simulated DTR over the northern





region. As for southern China, the RMSE of reconstructed SM can be reached 0.0342, 0.0353 and 0.0357 cm³/cm³, when

respectively using the other three dataset. Nevertheless, the availability of reconstructed SM products is remarkably increased (by ~6-11%) due to the all-weather coverage of reanalysis and land surface model simulations. This suggests the surface temperature source from the numerical model dataset can be one alternative for satellite LST, which is essential at one long-term and large extended scale, especially considering their full coverage attribution. On the other hand, as illustrated in Fig. 14 (b) and (c), one noticeable reduction in accuracy metric (~4%) occurs when not considering the variable

correction procedure. This emphasizes the indispensable contribution of variable calibration procedures in reconstructing surface characteristics (Duan and Bastiaanssen, 2013; Liu et al., 2020a).

Since the reanalysis SM is one vital input in our approach, we also compare it with the other two products to evaluate the feasibility of ERA data in reconstructing CCI SM. GLEAM and Noah surface SM are respectively employed to replace the ERA SM while other explanatory variables keep the rest the same. Although the GLEAM and Noah SM-based schemes can

demonstrate acceptable accuracies, they exhibit slightly inferior accuracies compared to ERA SM-based schemes (Fig.14 (b) and (c)), probably due to their relatively large uncertainties in delineating the surface soil moisture dynamics across the two selected regions (Mahto and Mishra, 2019; Chen and Yuan, 2020). Nevertheless, considering our study merely focuses on two local regions, we cannot manifest that the ERA product could provide the best performance across China. More attention should be focused on further work.

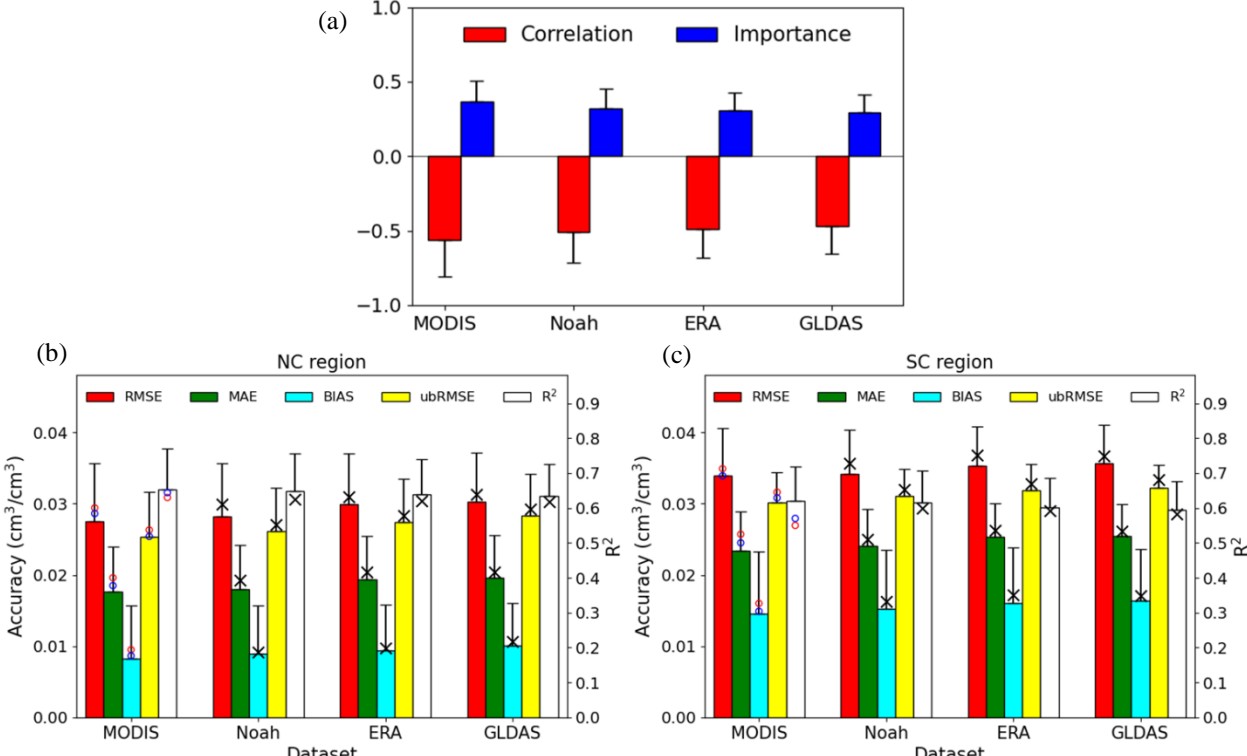






**Figure 14: The performance of models using Noah, ERA, and GLDAS DTR replacing MODIS DTR. (a) The Pearson correlation and importance score of DTRs. (b) and (c) The metrics of models using different DTRs for Northern China (NC) and Southern China (SC), respectively. Error bars denote 1σ errors. The symbol 'x' represents the accuracy metrics of the models without DTR correction procedure. The symbol 'o' in red represents the accuracy metrics of the models using GLEAM SM to replace ERA SM, and the symbol 'o' in blue represents the accuracy of the models using Noah SM to replace ERA SM.**

### 4.6 Extending to one long-term scale

The available dataset forcing our model has a long sequence, implying one potential in modeling long-term SM products. To verify this, the proposed gap-filling method is further extended to the long-term ECA CCI SM databases. During 2005—2015, more than 90% of the contaminated pixels can be reconstructed using our model, as illustrated in Fig. 15(a) and (b). When evaluating the pixels against dense in situ measurements in the Maqu network, we observe that the reconstructed SM has comparable accuracy with the original SM (Table 5). The average $R^2$ and RMSE of reconstructed SM are 0.73 and 0.12 cm$^3$/cm$^3$, respectively. The present results indicate the proposed model has a strong capacity to simulate SM at a long-time scale.

Figure 15(c) shows obvious differences between the gap-filled and original SM dataset. Negative differences in SM occur in most regions, while positive differences happen in a small fraction of the wet and arid regions. The dynamic and trend of SM are fundamental to assessing and quantifying eco-hydrology regime. Accordingly, we further investigate the trend of SM series during 2005-2015, which is obtained with Sen's slope and M-K significant analysis (Li et al., 2021b; Liu et al., 2021a). As shown in Fig. 15(d)-(f), the difference in valid SM values participating in trend analysis brings about a noticeable disparity in SM trend, which implies a slightly decreased SM trend for most arid regions in China and an increased SM trend for most humid regions. This pattern in SM trend is mostly related to the climate changes (precipitation and temperature) changes and human activities (Li et al., 2018b).

The biases in SM dynamic and trend are more pronouncedly delineated for each climate region in Fig. 16(a) and (b). Results show that the trends from the reconstructed SM are relatively lower compared to those from the original CCI SM. The improvement of the reconstructed dataset in depicting SM trends are quantitatively manifested in Fig. 16(c) and (d), which demonstrates the $R^2$ between the trends from the original CCI SM and those from in situ measurement is 0.23 while the $R^2$ between the trends from the reconstructed CCI SM and those from observations is increased to 0.45. Overall, an effective gap-filled model is demanded considering its capacity in fully depicting dynamics and trends of SM.

**Table 5 Metrics for the gap-filling performance regarding Maqu network for the extended years**

| Year | Metrics for raw CCI dataset | | | | | | Metrics for gap-filled dataset | | | | | |
|------|------|------|------|------|--------|------|------|------|------|------|--------|------|
|      | $R^2$ | RMSE | MAE | Bias | ubRMSE | NSE | $R^2$ | RMSE | MAE | Bias | ubRMSE | NSE |
| 2008 | 0.8  | 0.11 | 0.1  | 0.06 | 0.06   | 0.8  | 0.71 | 0.13 | 0.13 | 0.07 | 0.06   | 0.81 |
| 2010 | 0.82 | 0.1  | 0.09 | 0.05 | 0.06   | 0.81 | 0.73 | 0.11 | 0.11 | 0.06 | 0.05   | 0.83 |
| 2011 | 0.83 | 0.09 | 0.09 | 0.06 | 0.06   | 0.82 | 0.74 | 0.11 | 0.1  | 0.06 | 0.05   | 0.84 |
| 2012 | 0.81 | 0.12 | 0.09 | 0.06 | 0.05   | 0.81 | 0.72 | 0.13 | 0.12 | 0.05 | 0.05   | 0.82 |
| 2013 | 0.82 | 0.09 | 0.09 | 0.06 | 0.05   | 0.8  | 0.73 | 0.12 | 0.13 | 0.07 | 0.07   | 0.82 |



| 2014 | 0.85 | 0.09 | 0.08 | 0.06 | 0.05 | 0.83 | 0.74 | 0.11 | 0.09 | 0.08 | 0.06 | 0.85 |
| 2015 | 0.79 | 0.12 | 0.1 | 0.07 | 0.07 | 0.79 | 0.69 | 0.14 | 0.12 | 0.09 | 0.07 | 0.81 |

Note: NSE is from the evaluation with the time series of average 0.25º pixels while the other five metrics are from the evaluation with 1 km disaggregated values.

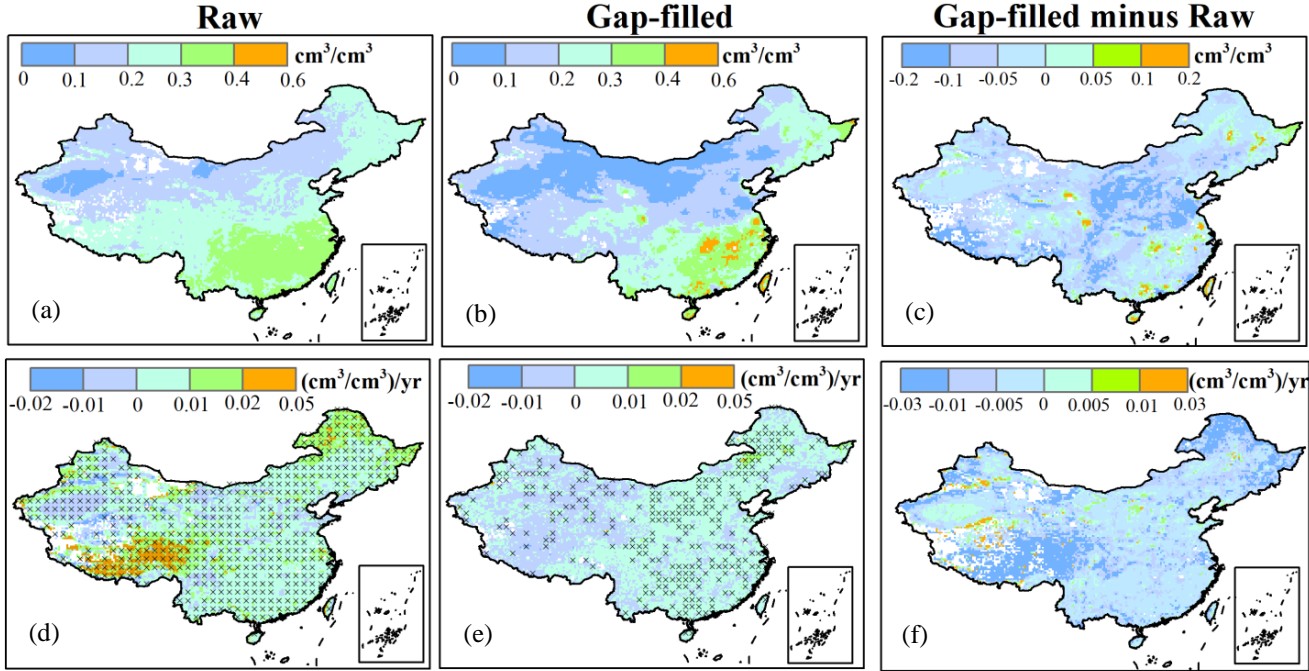

**Figure 15: The implementation of the proposed model to 2005-2015. (a) and (b) are the average values of raw CCI and gap-filled SM during 2005-2015, and (c) is the difference between them. (d) and (e) are the average trends of raw CCI and gap-filled SM during 2005-2015, and (f) is the difference between them. The symbol "x" in (d) and (e) denotes the significance level under 0.05.**



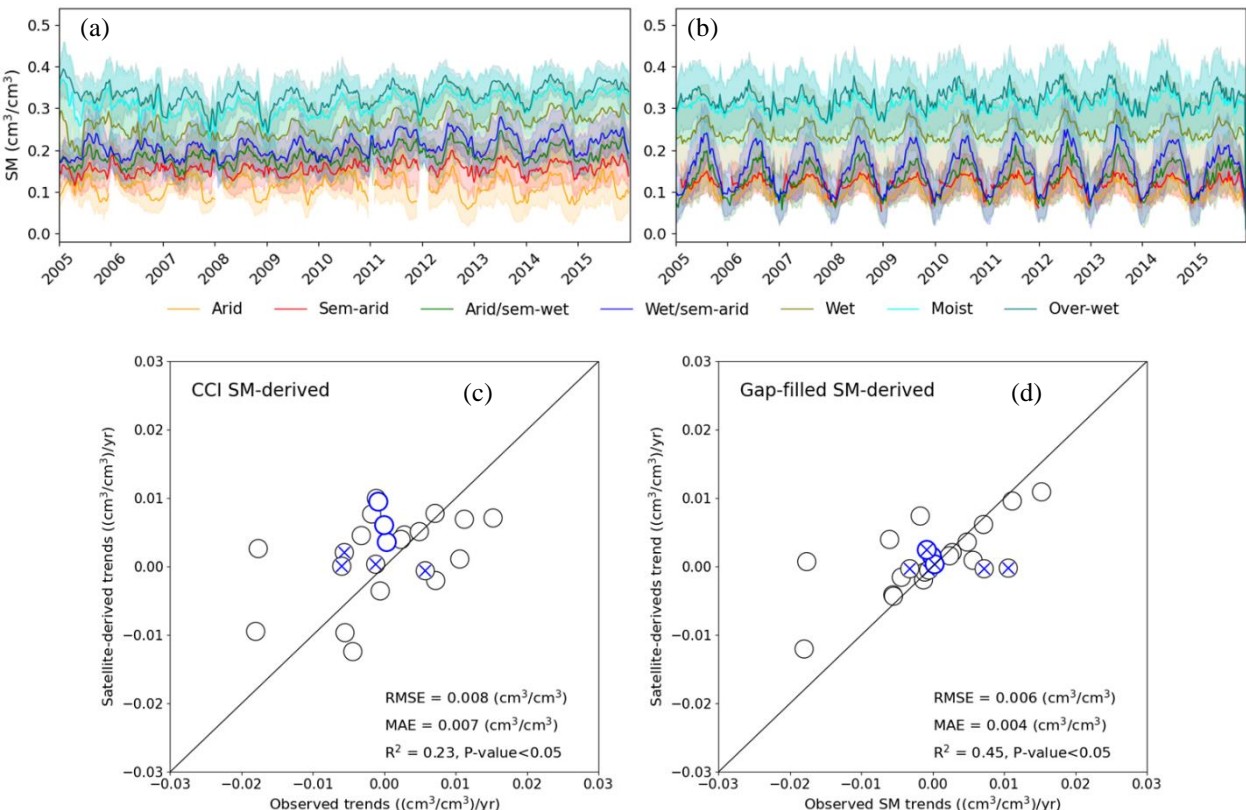

**Figure 16: (a) shows the temporal patterns of raw CCI SM regarding different climate regions during 2005-2015, and (b) shows the temporal patterns of gap-filled CCI SM regarding different climate regions. The shaded area in (a) and (b) denotes ±1 standard error. (c) The scatter plot of 1-km CCI SM-derived trends against in situ measures during 2005-2014, and (d) the scatter plot of 1-km gap-filled SM-derived trends against in situ measures. The blue circle in (c) and (d) means the trends from in situ measures are under insignificance level, while the fork means the trends from satellite-derived SM are under insignificance level.**

## 5. Conclusions and future considerations

The continuity of satellite-derived SM series is hampered by the data gap issues. This study thus provides a novel framework for reconstructing a spatially continuous daily SM dataset by integrating the ESA CCI SM and the related explanatory variables. To achieve this, one random forest method taking full account of both the spatial and temporal domains is designed. The explanatory variables filtered on the basis of a spatiotemporal window search strategy exhibit a substantial effect in driving RF regression, resulting in an efficacy improvement of ~19%. Meanwhile, model performance is enhanced by calibrating the derived residuals based on GWR and Gaussian filters. This improvement is manifested by the fact that the accuracies of gap-filling models are lowered by ~9% when removing the residual calibration procedure.

Study presents the merit of identifying a sufficient number of explanatory variables from the integration of satellite observations and model-driven knowledge. The selected variables complementarily reproduce the SM dynamics in addition to capturing the spatial variations, which also implies that the nonlinear correlation between the SM and explanatory





variables can be delineated on the spatiotemporal scale. Importance analysis illustrates that the accuracy of reconstructed SM

is noticeably reduced when excluding each participating variable while keeping the rest variables. Specifically, in addition to conventional variables from optical remote sensing, the essential environment elements from model-driven knowledge are used to improve the performance of SM reconstruction. Earlier studies have suggested (Li et al., 2021a; Long et al., 2019) that the reanalysis dataset and land surface model product can provide spatiotemporally continuous records, indicating the great potential of simulating land surface parameters. Our study proposes to merge CCI SM time series with the reanalysis

and land surface model dataset and applies to China. The reconstructed SM achieves satisfying accuracy, especially for areas with large swath gaps, underscoring the importance of spatial coverage and continuity of the environmental factors, and the multiple datasets should be involved in the gap-filled models. We further confirm this with one uncertainty analysis showing the feasibility of using alternative data sources of DTR and SM, which is essential at one long-term scale, especially considering the full coverage attribution of numerical model simulated products. Nevertheless, since the numerical model

simulated models are generally sensitive to regional surface and climatic conditions, more representative model products such as CLDAS and regional numerical models can be considered in further work.

Machine learning is previously reported to be a powerful strategy for reconstructing contaminated values. Despite the effectiveness of RF models in situ SM databases, its applicability in reconstructing long-term satellite observational records especially across a large scale still deserves careful investigation. Here we manifest that the random forest, combined with

the appropriate covariate exploiting both the spatial and temporal domains and the model-derived residual calibration module, can be a robust method in gap-filling the CCI SM database over China. The superiority of RF-based model in reconstructing SM is further proved by comparing it against the other four models. Despite this, more advanced machine learning strategies, such as deep neural networks (DNN) and long short-term memory (LSTM), are expected to enhance simulation accuracy. Specifically, the ensemble approaches mainly accounting for the scale biases among different gridded

dataset are required. For example, one Bayesian modeling framework that can provide simulation standard error using uncertainty quantification is encouraged (Zhao et al., 2019a).

The variables forcing the proposed model are all from reliable data and long-term worldwide. Accordingly, the proposed method can be extended to generate a promising long-term gap-filled SM dataset. This is critical considering the spatiotemporally continuous SM is demanded for ecological and hydrological research. A promising result with a $R^2$ of 0.72

is observed when applying our gap-filling approach to long-term SM data sets (2005-2015) in China. In particular, a more accurate trend is achieved relative to that of the original CCI SM when assessed with in situ measurements (0.45 versus 0.23 in terms of $R^2$). Overall, our study may provide several insights for continuous monitoring of surface water dynamics and drought, and further promote the research of water resources management and climate change.



**Code/Data availability**

All the datasets used in this study are open to the public. The National Aeronautics and Space Administration team provides the MODIS products, SRTM DEM data and GLDAS data freely download via the website https://earthdata.nasa.gov/. The ESA CCI soil moisture dataset and ERA-5 reanalysis datasets is collected from the European Centre for Medium-range Weather Forecasts (ECMWF) for providing (https://www.ecmwf.int/en/forecasts/datasets). The Brecht Martens, Diego Miralles and their team provides the GLEAM datasets (http://www.gleam.eu/). The China Watershed Allied Telemetry

Experimental Research (WATER) project, Chinese Ecosystem Research Network (CERN), and Maqu soil moisture monitoring network provides available in situ measurements at the website http://data.tpdc.ac.cn/en/. The Chinese regional ground meteorological dataset is collected from the National Tibetan Plateau Data Center (http://data.tpdc.ac.cn).

**Author contribution**

Kai Liu, Xueke Li, and Shudong Wang designed the theoretical formalism. Kai Liu performed the analytic calculations.

Both Shudong Wang and Hongyan Zhang contributed to the final version of the paper.

**Competing interests**

The contact author has declared that neither they nor their co-authors have any competing interests.

**Acknowledgments**

This study was jointly supported by the Natural Science Foundation of China (42141007 and 41671362), and the Inner

Mongolia Autonomous Region Science and Technology Achievement Transformation Special Fund Project (2021CG0045).

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
