# Peer review of "A robust gap-filling approach for ESA CCI soil moisture by integrating satellite observations, model-driven knowledge and spatiotemporal machine learning"

_Hydrology and Earth System Sciences, 2022_

## Author Comment (AC1)

The study presents a method for gap-filling ESA CCI soil moisture data. For filling the gaps, the approach utilizes information from the spatiotemporal domain around the missing value as well as from other explanatory variables. This is a timely contribution to the ever-growing field of gap-filling and data fusion in Earth system science. The study benchmarks the method over China, which covers a large variety of different climate zones and topography, making it a suitable test bed. Particularly with the severity of missing values in microwave remote sensing soil moisture retrievals, it is vital to use as much information as possible, both from the spatial and temporal domain as well as from other available observations. Therefore, the proposed method seems promising to be able to do the task. Furthermore, comparing the gap filled values to in-situ observations gives an independent insight into gap filling performance.

However, the poor use of language and grammar hinder understanding of the methods and results in many cases. Additionally, the structure of the results and methods sections are a bit unclear and seem arbitrary, such that following the storyline of the paper is difficult. Finally, the purpose, aim and implementation of some of the methodological choices and evaluations in the result section are unclear and should either be replaced or clarified.

I therefore think the paper requires major revisions. Please see below my general and specific comments.

Response: Thanks for your valuable time and constructive comments. We tried our best to incorporate them in the new version. Especially, the English language usage in this version will be improved deeply by one fluent English researcher that is suggested by the commercial corporation.

GENERAL COMMENTS

1. The study needs restructuring especially in the methods and results section. It is hard to follow the methods and results section, since they are structured very differently. Please align Figure 2 with the structure of Section 3, such that it is easy for the reader to follow which part of the model is described. For example, the subsections could have the same header names as the boxes in Figure 2. Also, it is not clear why only parts of Figure 2 are described in the subsections. Furthermore within the results, sometimes two different results are explained in one subsection (e.g. Section 4.2. compares to in-situ measurements and to the "cross-validation", Section 4.3 compares to literature and to other employed methods). I suggest making individual sections for individual results, or clarify why the results are thrown together in one Section like this.

Response: Thanks for pointing out this issue. In the revised version, the Figure 2 and the related context will be rewritten to make it align with the structure of section 3. Meanwhile, the Section 3 and 4 is reorganized, in particular, the subsection in these two sections are rearranged to make it clear and readable.

[Figure]

Figure 2: The overall diagram of the proposed methodology. The main work carried out includes dataset processing, model implementing, and model analyzing. The red text denotes the core procedures conducted in the proposed model, which will be further described in the following sections.

2. In general within the result section, the results are described with generic terms like "good coindicence", "capacity for reconstruction", "high accuracies", "strong variation delineating capacity". Whilst those all are incorrect English terms, they also do not go into detail or describe the results well. While it is important to give statistical evidence of the algorithm performance, it is of equal importance to discuss the difference between the statistical measures and possible reasons for it, as well as evaluating the physical plausibility and coherency of the gap filled values. It would highly benefit the study if the results were described more concisely, more descriptive words would be used, and the results would be discussed, taken into context and described better. A good example of a good explanation of results is for example found in L433-436. Please more of those. Furthermore, it would be beneficial to add at least one plot that evaluates the physical plausibility of the gap-filled values for a possible application of the gap filled dataset. As an example, since soil moisture values are often important when analysing droughts, it would be interesting to see whether the gap-filling method is able to not only work in the mean of the values, which all the statistical metrics aim at, but also see whether the extreme values are gap-filled well. This could be done by showing some maps of a known drought event over China, possibly compare with the in-situ measurements and the original, gappy CCI SM data.

Response: Thanks for the reviewer's suggestion. In the new version, the result section will be revised and more descriptive words are added to make it more convinced and concise. Meanwhile, to reveal the physical plausibility of gap-filled soil moisture, we pay particular attention to the evaluation of gap-filling SM under extremely dry conditions. The extreme drought is defined based on meteorological condition, i.e., the Palmer Drought Severity Index (PDSI) is less than −2 over 8 consecutive months or longer (Fig.

S2). Specifically, 1) we will add the accuracy evaluation in Fig. 7 and 8 focusing on the drought regions; 2) we will label the drought regions in Fig. 9 to make it more readable; and 3) we will analyze two typical drought regions by comparing the difference source SM (Fig. S4).

[Figure]

Figure 7: The evaluations of model results. (a), (c) and (e) are the scatter plots of 1-km CCI SM-derived values against field measures regarding WATER/CEERN, agro-meteorological stations, and Mauqu network, respectively, and (b), (d) and (f) are the scatter plots of 1-km gap-filled SM-derived values against field measures. The sub-figures in the lower corner of (a)-(d) are the scatter plots under extremely dry conditions. (g) are the time series of average CCI SM-derived values against site measures in the Maqu region. The shaded area in (g) denotes ±1 standard error.

[Figure]

Figure 8: The accuracy metrics of 10-cross validation for R2, RMSE, MAE, BIAS, ubRMSE and NSE: (a) is averagely obtained on a month basis, and (b) is averagely obtained for each climate region.

[Figure]

Figure 9: The spatial distributions of accuracy metrics of 10-cross validation in 2009 for R2, MAE, BIAS, ubRMSE and NSE. The slash represents the regions impacted by drought.

[Figure]

[Figure]

Figure S2. (a) The annual PDSI in 2009. (b) The spatial distribution of drought events, and two severe drought events (D1 and D2) selected for further analysis.

[Figure]

Figure S4. The time series in the (a) region D1 and (b) D2. D1 and D2 are identified in Figure S2.

3. Throughout the script, there are many spelling and grammar mistakes, some of which significantly hamper understanding of particular methodological decisions or results. It is vital that these mistakes, some but not all of which are listed below, should be removed from the script. A thorough cleanup of the language and individual sentences is necessary. Many statements could also be reduced, whilst keeping the meaning, to a minimum of words, effectively improving the structure and readability of the script and at the same time reducing its size.

Response: Thanks for pointing out this issue. In the new version, the English language usage will be improved deeply by one fluent English researcher that is suggested by the commercial corporation. We also will reduce some statements to improve the structure and readability of the script.

4. It is a reasonable and thoughtful decision to investigate the importance of the different selected variables for the gap-filling process. However, I am not yet fully sure whether the selected method is the best in this context and whether it is applied correctly. Firstly, the mentioned model produces a

significance score that I cannot assess, since the equation is not provided. It would be vital to add this, since only then the meaning of the significance (e.g. in Fig 3a) can be understood. Furthermore, the cited literature uses this method only for exploratory variable importance in hydrological case studies and not to limit variables that feed a gap-filling model. In the latter case I suggest another method that is well-known and often used in the machine learning context for feature selection prior to model employment could be used. These methods include the feature importance assessment of the Random Forest model, a permutation feature importance assessment, introducing a regularised regression that puts weights on the input features, or evaluating using the SHAP value. Please provide in your answer to this review a discussion on which method you settled and why you chose this one. Furthermore, the used method is a linear regression, which screens for important variables only in linear relationships to soil moisture. However, the relationship between soil moisture and the provided explanatory variables is in many cases non-linear, therefore there is a chance that important dependencies are missed if only for a linear relationship is screened. Additionally, the missing regularisation in the linear regression makes it prone to overfitting. Finally, it is unclear to me why a variable selection / importance assessment is conducted twice, in Section 3.1 and in Section 4.4. I suggest only making one of those assessments, e.g. by removing Section 4.4. and potentially replacing the assessment in Section 3.1 with a more robust method, see my suggestions above.

Response: Thanks for the reviewer's valuable suggestion.

- Currently, there is no consensus on the optimal model to select explanatory variables considering the complexity of hydrological scenarios. For example, RF analysis can indicate plausible governing processes from emergent relationships, but by construction it does not suggest causality. One of our study aims is to provide one general framework for gap-fill soil moisture that can be extended to other machine learning approaches. Accordingly, one general method, Pearson correlation, is used to select the explanatory variables. It should be mentioned that the Parson correlation have been extensively used in delineating soil moisture especially considering it can depict some specific non-linear properties, e.g., lag time when combing with step strategies [1]. In this version, according to the reviewer's suggestion, details descriptions about the factors selection approach will be provided in the supplementary Text S1.

Text S1: The regression subset selection approach

The main assumption beneath this regression subset selection approach is that the suppressor variables are associated significantly with each other in regression models, although they may be less correlated with the dependent variables. To be specific, this approach can be conducted with the following steps: (1) using least-squares linear regression to check the potential relationships between SM and explanatory variables; (2) applying a backward stepwise (remove) regression to explore the potential explanatory

variables based on the Akaike Information Criterion (AIC); (3) exploiting the best models from all variable combination to identify the important variables impacting SM; and (4) quantifying the relative contributions of each explanatory variable to SM based on the determination coefficient.

- To validate results of Pearson correlation, according to the reviewer's suggestion, we also employ permutation feature importance which measures the relative importance of each predictor variable from the difference of errors before and after a temporal permutation applied to the particular variable. Similar results (in Figure 3(a)) as those from Pearson correlation illustrate the robust importance-based analyses we used. In addition, the importance score produced by RF will be removed to supplementary.

[Figure]

Figure 3: The significance percentage and permutation importance of the selected variables in correlation to CCI SM.

[1] Almendra-Martín, Laura, et al. "Comparison of gap-filling techniques applied to the CCI soil moisture database in Southern Europe." Remote Sensing of Environment 258 (2021): 112377.

5. Why do you use a combination of scaling algorithms in Section 3.1.2? Since you never compare ESA and ERA soil moisture values in the results, you could just compute standardised anomalies of the ESA values, run the gap filling model and then add mean and standard deviation again, to convert back to physical values? Please explain, in case I have missed why this step is necessary, or just use standardised anomalies for simplification.

Response: Although the reanalyses dataset could reproduce observed characteristics have improved significantly over the years, its performance might be limited when there are some systematic errors in reanalysis products [1]. Systematic biases are unavoidable in global reanalyses, and these biases can be

propagated through the simulation process (e.g., gap-filling and downscaling) with far reaching implications on the simulation products and any subsequent applications [2]. Accordingly, bias correction of reanalyses prior to dynamical simulation is required to ensure that errors are corrected, leading to better skill and consistent simulated products that are independent of the choice of the domain.

Our study is also hoped to verify the ability of bias correction of reanalysis or model simulations prior to gap-filling procedure for soil moisture reconstruction, especially our study is focused on the reanalysis or model simulations. Running the model skimpily using the standardized anomalies might be unfit for some variables that own complex characteristics such as the LST.

In the new version, we will revise the related context to make it clearer. Meanwhile, the histograms are provided in supplementary Fig. S1 to illustrate the differences in soil moisture between ESA and ERA.

[Figure]

Figure S1. The spatial distributions of ESA CCI SM, ERA5 SM and calibrated ERA SM on the selected days of 2009. The lower-left panel in each sub-figure shows the histogram, and the blue color represents the pixels in which the ESA dataset are available while the red color represents the pixels in which the ERA dataset are available.

[1] Moalafhi D B, Sharma A, Evans J P, et al. Impact of bias-corrected reanalysis-derived lateral boundary conditions on WRF simulations. Journal of Advances in Modeling Earth Systems, 2017, 9(4): 1828-1846.

[2] Mehrotra R, Sharma A. Correcting for systematic biases in multiple raw GCM variables across a range of timescales[J]. Journal of Hydrology, 2015, 520: 214-223.

6. In result Section 4.1 you compare the original daily CCI SM and the corresponding gap-filled dataset for the year 2009. You argue that the model performance has merit based on the fact that the two datasets show similar characteristics in Figure 6. This however is not an argument that is relevant in gap-filling. Since the missing values in the original CCI SM data are missing not (completely) at random, but are missing systematically where vegetation cover is high or the soil is frozen, this dataset is biased towards the underlying, unobservable gap-free "truth". Comparing the original, gappy data with the gap-filled that is supposed to produce biases, i.e. change the statistical moments of the data because it is missing not at random (see e.g. Rubin et al, 1976, Little et al 2014, Van Buuren et al 2018, Bessenbacher et al 2022). If these two datasets are compared, it should be aimed towards physical plausibility and coherency, and not about whether they are statistically similar.

Response: Thanks for pointing out this issue. In the new version, we will add the related context to describe the bias issue. Meanwhile, the histograms of two dataset are compared to explore the value distribution properties of different products. In addition, the Nash-Sutcliffe Efficiency (NSE) is also added to measure the overall performance of the proposed model, as illustrated in Fig. 8 and 9.

[Figure]

Figure 6: The spatial distributions and histogram of the raw and gap-filled CCI SM on the 15th of each month in 2009.

[Figure]

Figure 8: The accuracy metrics of 10-cross validation for R2, RMSE, MAE, BIAS, ubRMSE and NSE: (a) is averagely obtained on a month basis, and (b) is averagely obtained for each climate region.

[Figure]

Figure 9: The spatial distributions of accuracy metrics of 10-cross validation in 2009 for R2, RMSE, MAE, BIAS, ubRMSE and NSE. The slash represents the regions impacted by drought.

7. It is unclear why a cross-validation is performed on presumably the year 2009. L335 only cites generically "model performance" as a reason. A cross-validation is usually performed to find the corresponding parameter values of the model (here the Random Forest), but this procedure is already described in Section 3.2 and refers to the years 2003-2008 which are never shown in the result section. Which parameters are defined with the cross-validation as described in L334-335? Which values are Figures 10 and 11 comparing to? The purpose and aim of this part of the results section is unclear to me.

Response: Evaluating the gap-filled SM with in situ stations is supposed to produce biases that can be caused by scale mismatching and disaggregation model performance. To account for this, one holdout cross-validation with ten replicates is performed in 2009 to comprehensively evaluate the model performance. For each replicate, we randomly hold out 10% of pixels, i.e., introducing the manual gaps for these pixels, and train the model with the remaining 90% of the dataset. After the gap-filled SM series of held-out pixels are reconstructed from the training set, they will be validated with the original SM. This strategy has been extensively used for evaluating the accuracy of the gap-filled data, considering its merits in overcoming the systematic errors and scale mismatches between the reconstructed values and field measures [1-3]. In the new version, we will add the related context to make it clearer.

[1] Almendra-Martín L, Martínez-Fernández J, Piles M, et al. Comparison of gap-filling techniques applied to the CCI soil moisture database in Southern Europe. Remote Sensing of Environment, 2021, 258: 112377.

[2] Zhang T, Zhou Y, Zhu Z, et al. A global seamless 1 km resolution daily land surface temperature dataset (2003–2020). Earth System Science Data, 2022, 14(2): 651-664.

[3] Meng X, Liu C, Zhang L, et al. Estimating PM2. 5 concentrations in Northeastern China with full spatiotemporal coverage, 2005–2016. Remote sensing of environment, 2021, 253: 112203.

8. Section 4.5: It is unclear why suddenly focus regions are introduced and used, and why this analysis cannot be conducted on the whole area (all China). Also, it is unclear why the focus regions are so different in size. This makes them harder to compare, as the wet region has less datapoints to compare and covers a much less diverse climate zone (compare Figure 1). Please clarify why these decisions are necessary or make the analysis on the whole of China. Furthermore within this section, please verify that none of the SM data from ESA, GLEAM and Noah is used for ERA. If that would be the case, the datasets are not independent, and this could for example explain that they are more similar. Similarly, check that neither MODIS or GLDAS are used for Noah or ERA runs.

Response: Thanks for pointing out this issue. This section focuses on the uncertainty regarding LST and reanalysis SM. Despite the satellite LST is closely related to surface soil moisture, it is also impacted by

contamination issues. The reanalysis dataset and outputs from land surface model can provide the full coverage data, but is accuracy is generally impacted by the numerical model used. Accordingly, the uncertainty analysis is should be focused by investigating the substitution performance of other surface temperature sources in reconstructing SM.

Noah soil moisture and LST is introduced to conduct uncertainty analysis. In our study, this model is run with regional forcing variables rather than the global parameterizations, and thus could provide more accurate soil moisture and LST. Moreover, the utilization of regional model simulations can provide references for dataset gap-filling with other regional reanalysis dataset. The two focused regions are selected because sufficient soil moisture and surface can be provided by Noah model from our previous studies [1, 2]. Here we merely used these regions for uncertainty analysis rather than the comprehensive comparison. What's more, it is not cost-effective to run the Noah model for the whole China.

As far as I know, ERA is a reanalysis dataset that is forced by reanalysis meteorological dataset. The soil moisture of ERA doesn't depend on the products from ESA, GLEAM and Noah [3]. The noticeable differences among them have been illustrated in previous studies [4, 5]. Meanwhile, the surface temperature of all the available dataset is not depended.

According to the reviewer's suggestion, in the new version, we will add the related context to make it clearer.

1] Liu K, Su H, Li X, et al. Development of a 250-m Downscaled Land Surface Temperature Data Set and Its Application to Improving Remotely Sensed Evapotranspiration Over Large Landscapes in Northern China. IEEE Transactions on Geoscience and Remote Sensing, 2020, 60: 1-12.

[2] Liu K, Li X, Wang S. Characterizing the spatiotemporal response of runoff to impervious surface dynamics across three highly urbanized cities in southern China from 2000 to 2017. International Journal of Applied Earth Observation and Geoinformation, 2021, 100: 102331.

[3] Muñoz-Sabater J, Dutra E, Agustí-Panareda A, et al. ERA5-Land: A state-of-the-art global reanalysis dataset for land applications. Earth System Science Data, 2021, 13(9): 4349-4383.

[4] Ling X, Huang Y, Guo W, et al. Comprehensive evaluation of satellite-based and reanalysis soil moisture products using in situ observations over China. Hydrology and Earth System Sciences, 2021, 25(7): 4209-4229.

[5] Wu Z, Feng H, He H, et al. Evaluation of soil moisture climatology and anomaly components derived from ERA5-land and GLDAS-2.1 in China. Water Resources Management, 2021, 35(2): 629-643.

9. Since this gap-filling method could be an important contribution to the problem of missing values in soil moisture remote sensing observation and Earth observations in general, it would be beneficial if the code of the method could be published, preferably on a platform that enables easy use for interested users and a versioning system (e.g. Github). Also for the purpose of this review, if possible, it would be interesting to have a look at the code to understand better what is going on and how exactly this is implemented.

Response: Thanks for pointing out this issue. Our work is primarily conducted in the Python with core packages including gdal, NumPy, scikit-learn and scipy, as well as in the RStudio with the package of spgwr. In the next step, we will reorganize the python code in a friend manner and put it on the Github platform.

Regarding our work, two core codes mainly includes that i) for finding the searched windows (nw) and searched days (nd), and ii) for using the random forest to gap-fill object soil moisture. The core part is provided in the following text.

**Code #1 for finding the searched windows (nw) and searched days (nd)**

```
        nw = 4
        nd = 1
        n_sample = 0
        while (nw <= 10) & (n_sample < 8*num_variable) :
          sm_local = np.ravel(sm[iband, i-nw:i+nw, j-nw:j+nw])
          index = (sm_local > 0)
          n_sample = len(sm_local[index])
          nw = nw + 1
        nw = nw - 1

        if (n_sample < 8*num_variable):
          nd = nd+1
          while (nd <= 4) & (n_sample < 8*num_variable) :
            sm_local = np.ravel(sm[iband-nd:iband+nd, i-nw:i+nw, j-nw:j+nw])
            index = (sm_local > 0)
            n_sample = len(sm_local[index])
            nd = nd + 1
          nd = nd – 1
```

**Code #2 for using the rf to gap-fill object sm**

```
            sm_rg = sm_local[index]
            albedo_rg = albedo_local[index]
            ndvi_rg = ndvi_local[index]
            pet_rg = pet_local[index]
            ap_rg = qp_local[index]
            era_sm_rg = era_sm_local[index]
            tpi_rg = tpo_local[index]
            dtr_rg = dtr_local[index]

            explain_value = np.vstack([albedo_rg,ndvi_rg,pet_rg,ap_rg,era_sm_rg,tpi_rg,dtr_rg])
            rfr = RandomForestRegressor(n_estimators=estimators[itype], max_depth=depth[itype],
min_samples_split=split[itype],max_features=feature[itype])
            rfr.fit(explain_value.T, sm_rg)
            pred_value=
np.vstack([albedo[iband,i,j],ndvi[iband,i,j],pet[iband,i,j],ap[iband,i,j],era_sm[iband,i,j],tpi[iband,i,j],dtr[iband,i,j]])
            value_temp = rfr.predict(pred_value.T)
            if (value_temp > 0) and (value_temp < 1):
             sm_fill[iband,i,j] = value_temp
```

**SPECIFIC COMMENTS + TECHNICAL CORRECTIONS**

1. L20 "Compared to that…" I don't understand this sentence. Please clarify

Response: Thanks for pointing out this. We will revise this in the new version.

2. l31: "SM has been declared" please add citation

Response: This will be revised in the new version.

3. L44 there is some literature on the shortcomings of soil moisture assimilation into reanalysis, see e.g. Dorigo et al 2017

Response: Thanks for pointing out this issue. The related references will be added in this version.

4. L62: "some studies" but only one study is cited.

Response: Some other related references will be added in this version.

5. L74 and all other occurrences throughout the text: the word "delineating" is used in an unusual way. I suggest replacing it with depict/represent/show/ or similar in all occurrences

Response: Thanks for pointing out this issue. We will solve this issue in the new version.

6. L89 and all other occurrences throughout the text: using "the" in front of an previously unmentioned fact is confusing. Replace with "a" if not referring to a specific one.

Response: Thanks for pointing out this issue. We will revise this in the new version.

7. L95 consider citing Bessenbacher et al 2022

Response: The related references will be added in this version.

8. Fig 1: please increase resolution. Explain acronym DEM at first occurrence.

Response: Thanks for pointing out this issue. The quality of Fig 1 will be improved in the new version.

9. L119 "mainly". What is not included in this list?

Response: We will revise this in the new version.

10. Table 1: remove lines below "model analysis" in first column. Left-aligning columns could improve readability

Response: Thanks for pointing out this issue. We will solve this in the new version.

11. L 121: mention already here what the difference between "model establishment" and "model analysis" is. Is one the features used to run the model, and the other the evaluation? Not clear.

Response: Thanks for pointing out this issue. We will revise this in the new version.

12. L130: data is a plural word. Always "data are"

Response: We will double check this throughout the manuscript.

13. L142: to better understand the negative correlation of TPI and CCI SM in Figure 3 please add a short sentence explaining what a high / low TPI mean.

Response: Thanks for pointing out this issue. We will revise this sentence to make it cleaner.

14. L159: this is a good example of how the text can easily be shortened without losing understanding. "Precipitation, air temperature, … are obtained from the Chinese …". Please look for those sentences elsewhere as well to clear up text.

Response: Thanks for pointing out this issue. We will double check this throughout the manuscript.

15. L160: since this is dataset from ground stations, but you state it is gridded, how was the gridding procedure performed? Is there literature that you could cite?

Response: The dataset was generated through fusion of in-situ station data, remote sensing products and reanalysis datasets, and it has a temporal resolution of 3-hourly and a spatial resolution of 0.1° (He et al., 2020). We will revise this sentence in the new version.

16. Table 2: Since you never discuss results at the individual stations of the WATER and CERN stations, I don't think it is necessary to add a table here naming them all. Consider moving to Supplementary Material or adjust such that table only includes networks and not individual stations.

Response: Thanks for pointing out this issue. In the new version, the Table 2 is moved to Supplementary Material.

17. L184 please add citation for this claim.

Response: The related citations will be added for this claim regarding the station measurements.

18. L194 "a vector the sample number of which is decided by" no correct English sentence.

Response: Thanks for pointing out this issue. We will revise this sentence in the new version.

19. L197ff: I don't understand how steps (i) through (iii) are related to the four boxes in Figure 2. Adjust Figure 2 such that it has the same structure as the text, or vice versa.

Response: Thanks for the reviewer's valuable suggestion. In the new version, we will adjust both the Figure 2 and the related text to make them consistent.

20. Fig 2: Explain colours and frame shapes. Explain which criterion is to be met in "data judging". Explain which explanatory variables are taken from "dataset preparing" to "dataset judging". Rename judging.

Response: In the new version, we will rewrite the caption to make it readable.

21. L214: "plenty" as in all possible combinations? Are you stepwise removing or adding variables, or are you trying all different combinations? Please clarify.

Response: The main assumption beneath this regression subset selection approach is that the suppressor variables are associated significantly with each other in regression models, although they may be less correlated with the dependent variables. To be specific, this approach can be conducted with the following steps: (1) using least-squares linear regression to check the potential relationships between SM and explanatory variables; (2) applying a backward stepwise (remove) regression to explore the potential explanatory variables based on the Akaike Information Criterion (AIC); (3) exploiting the best models from all variable combination to identify the important variables impacting SM; and (4) quantifying the relative contributions of each explanatory variable to SM based on the determination coefficient. According to the reviewer's suggestion, we have added the related description in the new version.

22. L217: please define "importance criterion"

Response: The "importance criterion" means the determination coefficient.

23. L223: what happens in the case that gaps are present in the variables?

Response: Regarding the gaps presented in these variables, we will not further considered in order to avoid introducing additional errors. We have claimed this in the new version.

24. L224: are slope, lat, Lon, aspect and wind removed or not? They are not visible in Figure 3b,c but their removal is not mentioned in the text.

Response: Considering EVI and LAI have closely correlated to NDVI, and air temperature has closely correlated to DTR, we exclude EVI, LAI and air temperature, the aspect, slope, wind, latitude and longitude, in the further model application. We have claimed this in the new version.

25. Eq4: what does subscript p1 mean? What does the "." And the " " "mean in the second equation? Isn't mu(SM_c1) the same as mu(SM_ESA (t_av))? If so, please simplify equations.

Response: Thanks for pointing out this issue. We have revised this in the new version.

26. L254f: please define the difference between "traditional regression-based methods" and "machine learning approaches", since both are regressions, machine learning models do not inherently come with uncertainty estimation (for example, a Random Forest model does not have intrinsically uncertainty estimation and you don't have it in this study) and it is not less likely to overfit with machine learning methods.

Response: We have rewritten this in the new version.

27. L260 "is feasible to add layer categories" please clarify.

Response: RF is a hierarchical tree diagram, which is based on a nonparametric strategy and offers the opportunity to add a diverse variety of data layers into the model (Breiman, 2001).

28. L293: please clarify how missing values in the explanatory variables are treated within this algorithm.

Response: Considering that a fraction of gaps occur in the satellite dataset (e.g., LST and albedo) and the optimal window may not exist, the maximum values of sw and nd are introduced to terminate this process. On the other hand, regarding the gaps presented in these selected variables, we will not further considered in order to avoid introducing additional errors.

29. Fig 5, caption: please define sw and nd for better readability

Response: Thanks for pointing out this issue. We have revised the related context. The results of the sensitive analysis regarding two maximum values, i.e., the size of the spatial window (sw) and the number of temporal days (nd), for terminating the searching process.

30. L308: please clarify "neglectful variables"

Response: This calibration procedure can potentially correct the bias resulting from neglectful variables such as those are excluded for model establishment (Zhu et al., 2012; Liu et al., 2020a).

31. L311: please introduce acronym GWR before first mention.

Response: The full name of GWR (i.e., geographically weighted regression) is added in the new version.

32. L319: typo "her"

Response: Thanks for pointing out this issue. We have corrected this in the new version.

33. Section 3.4: Is this just a linear regression, applied to the same time window approach as the Random Forest, applied on the residuals from the Random Forest model? Please clarify

Response: We have clarified this in the new version.

34. L322: Please show a plot that shows the relative contributions of the GWR interpolation and the Gaussian Filter smoothing in the reply to this review, as to see that the influence of the latter on the results is smaller than the one of the former.

Response: We also check the accuracy of models excluding the residual calibration procedure, which is one essential procedure for the proposed model. Results in Fig. 10 demonstrate the accuracies are lowered by ~9% when removing residual calibration, underscoring the importance of residual modulation in improving SM reconstructing.

35. L354, 355: "heavy missing issues", "relative minor conformity" please correct English and clarify meaning.

Response: Thanks for pointing out this issue. We have corrected this in the new version.

36. L356: "consistent pattern" please clarify. Are you arguing they are similar or they have consistent biases? If yes, which? Please describe the results more.

Response: Thanks for pointing out this issue. We have clarified this in the new version.

37. Fig7, 8: Please put the corresponding days next to each other to simplify comparison.

Response: According to the reviewer's suggestion, we have adjusted the Fig.7 and 8 in the new version.

38. L388: I disagree that the values are close to the 1:1 line, but I also think that this is hard to achieve given the spatial gap between point measurements and gridded measurements.

Response: Thanks for pointing out this issue. We have deleted this in the new version.

39. L391: please clarify sentence "in general.."

Response: This sentence is revised in the new version.

40. L399: NSE is not introduced in Section 3.5. Please add.

Response: Thanks for pointing out this issue. We have added the related context in this version.

41. L402: please clarify sentence "in general.."

Response: We have removed this word in the new version.

42. Fig9g: please add the fraction of missing values for each day (e.g. similar to precipitation bar plots) such that it can be evaluated how the gap-filling performs with little or many missing values.

Response: According to the reviewer's suggestion, we have added the fraction of missing values for each day.

[Figure]

43. L423 please explain mechanism better

Response: According to the reviewer's suggestion, we will add the related context in the new version.

44. L431: again please refrain from simply describing the results as "good" without further analysis (see also general comment above)

Response: Thanks for pointing out this issue. We will revise the related context in this version.

45. L442 typo "in suit"

Response: This is corrected in the new version.

46. L442 "satisfied performance" please correct English

Response: This is corrected in the new version.

47. L446 "severe missing issues" please correct English

Response: This is corrected in the new version.

48. L453 "future compared" please correct English

Response: This is corrected in the new version.

49. L465 since the 9% & 19% increase in accuracy stemming from residual calibration and the spatiotemporal domains, respectively, is an important result that you mention in the conclusions and in the abstract, it should be more clear in Figure 12. Maybe add a figure with accuracy change per change in the method?

Response: According to the reviewer's suggestion, we will redraw this figure.

50. Figure 12, 13, 14: R**2 doesn't have a unit (Accuracy, cm**3/cm**3). Please clarify. For example, add an optimal value to each score (0 for RMSE, 1 for R2) and sort the diagram after scores, not after methods, such that the scores can be directly compared.

Response: In the new version, the original Figure 12, 13 and 14 will be redrawn.

[Figure]

Figure 10: Comparison RF-based model with other models (i.e., MLR, XGB, SVM and ANN). Error bars denote 1σ errors. The symbol 'x' represents the accuracy metrics of models excluding the residual calibration procedure, and the symbol 'o' represents the accuracy metrics of the models that use the global regression rather than regional regression based on the spatiotemporal window searching strategy.

[Figure]

Figure 11: The accuracy of the models removing one variable, i.e., using other six variables in model regression. Error bars denote 1σ errors. The text denotes the relative percentage of the decreased accuracy in comparison to the baseline accuracy using all seven variables.

[Figure]

Figure 12: The metrics of models using different DTRs for (a) Northern China (NC) and (b) Southern China (SC). Error bars denote 1σ errors. The symbol 'x' represents the accuracy metrics of the models without DTR correction procedure. The symbol 'o' in red represents the accuracy metrics of the models using GLEAM SM to replace ERA SM, and the symbol 'o' in blue represents the accuracy of the models using Noah SM to replace ERA SM.

51. L514 please clarify where this result is visible. If mentioning percentages of chance in the text, they should be visible in the graphs directly and not from comparing different bar plots visually.

Response: We also check the accuracy metrics of the models without correction procedure. We found one reduction in accuracy metric (~4%) in comparison with those using the correction procedure. We have further clarified this in the new version.

52. L532 "has a long sequence" please correct English

Response: This has been corrected in the new version.

53. L534 "more than 90%…" this is not clearly visible in Figure 15

Response: Thanks for point out this issue. We will revised this sentence in the new version.

54. L535 "comparable accuracy". Some metrics are better, some are worse. Discuss differences and possible reasons!

Response: The accuracy evaluation during 2005-2015 (excluding 2009) is only conducted using the Maqu network, which demonstrates that the reconstructed SM during this period has an acceptable accuracy with the original SM (Table 4) which is comparable to those in 2009. We will revise the related context in the new version.

55. L545 Please explain the mechanism better.

Response: Thanks for point out this issue. We will revise the related context in the new version.

56. Table 5: please reorder such that the values are immediately comparable, e.g. the R2 columns next to each other etc

Response: Thanks for point out this issue. The original Table 5 has been adjusted in the new version.

57. Fig 15: use different colorscale for last column to improve readability. For example, blue to red, for "wetter" to "drier"

Response: Thanks for point out this issue. The original Figure 15 will be adjusted in the new version.

58. Fig 16: one plot per climate zone, not one plot per dataset, such that they are visibly comparable. Also, disaggregate into seasonality and interannual variability to further analyse if both characteristics of the soil moisture dataset are reproduced in the gap-filled version.

Response: Thanks for the reviewer's suggestion. We have adjusted the original Figure 16. Meanwhile, we also discomposed the time series into to trend-cycle and seasonal component using one Seasonal-Trend decomposition using LOESS (STL).

59. L576 "study presents" please correct English

Response: This has been corrected in the new version.

60. L585 "especially for areas with large swath gaps" this is not shown in the results but would be very interesting

Response: Thanks for point out this issue. We will revise this in the new version.

61. L594 "manifest" please correct English

 Response: This has been corrected in the new version.

62. L692 "reliable data" too generic

Response: This has been rewritten in the new version.

REFERENCES

Response: Thanks for the reviewer's suggestion. We will try our best to incorporate these references in the revised version.

Rubin, D. B.: Inference and missing data, Biometrika, 63, 581–592, 1976.

Little, R. J. A. and Rubin, D. B.: Missing Data in Experiments, in: Statistical Analysis with Missing Data, pp. 24–40, John Wiley & Sons, Ltd, https://doi.org/10.1002/9781119013563.ch2, 2014.

van Buuren, S.: Flexible Imputation of Missing Data, Second Edition, Chapman and Hall/CRC, Boca Raton, 2 edition edn., 2018.

Bessenbacher, V., Gudmundsson, L. And Seneviratne, S.I: CLIMFILL v0.9: A Framework for Intelligently Gap filling Earth Observations, GMD (in review), https://doi.org/10.5194/gmd-2021-164

Dorigo, W., Wagner, W., Albergel, C., Albrecht, F., Balsamo, G., Brocca, L., Chung, D., Ertl, M., Forkel, M., Gruber, A., Haas, E., Hamer, P. D., Hirschi, M., Ikonen, J., de Jeu, R., Kidd, R., Lahoz, W., Liu, Y. Y., Miralles, D., Mistelbauer, T., Nicolai-Shaw, N., Parinussa, R., Pratola, C., Reimer, C., van der Schalie, R., Seneviratne, S. I., Smolander, T., and Lecomte, P.: ESA CCI Soil Mois- ture for improved Earth system understanding: State-of-the art and future directions, Remote Sensing of Environment, 203, 185–215, https://doi.org/10.1016/j.rse.2017.07.001, 2017.

---

## Author Comment (AC2)

This manuscript explores a RF based approach to fill the spatial gaps in satellite SM observations. The paper is extensive and well organized. The literature review is also extensive, nevertheless it explores studies that are immediately similar to the study to a high degree (i.e. predicts SM from satellite sensors) that there might be some lessons learned in some related studies (i.e. predicts reanalysis SM from model outputs using observed predictors similar to those used in this study) that weren't discussed. Other specific comments are below.

Response: Thanks for your valuable time and constructive comments. We will try our best to incorporate them in the new revision. In the modified version, some references (see the last sections) will be enhanced, and some lessons learned in some related studies are added especially in the discussion and conclusion section. In addition, the English language usage in this version will be improved deeply by one fluent English researcher that is suggested by the commercial corporation.

1. L85 ElSaadani et al. 2021 "Assessment of a spatiotemporal deep learning approach for soil moisture prediction and filling the gaps in between soil moisture observations" discussed in detail the following issues that are relevant to this manuscript:

- Filling SM gaps saptio-temporally using a convlstm ML approach.
- The effect of the number of predictors and time step of prediction on the model performance.

In addition, Q Li et al "Improved daily SMAP satellite soil moisture prediction over China using deep learning model with transfer learning" built on the above work to apply it to SMAP observations while improving the ML convlstm accuracy. Please include the above references for completeness of literature review and lessons learned regarding the predictors and their effect on ML model performance.

Response: Thanks for pointing out this issue. We will enhance the references throughout the manuscript to make it more convincible.

2. Table 5 Please separate the two sides of the table properly

Response: Thanks for pointing out this version. The original Table 5 has been adjusted in the new version.

Table  Metrics for the gap-filling performance regarding Maqu network for the extended years

| Year | $R^2$ | | RMSE (cm³/cm³) | | MAE (cm³/cm³) | | Bias (cm³/cm³) | | ubRMSE (cm³/cm³) | | NSE | |
|------|-----|-----------|-----|-----------|-----|-----------|-----|-----------|-----|-----------|-----|-----------|
| | CCI | gap-filled | CCI | gap-filled | CCI | gap-filled | CCI | gap-filled | CCI | gap-filled | CCI | gap-filled |
| 2008 | 0.8 | 0.71 | 0.11 | 0.13 | 0.1 | 0.13 | 0.06 | 0.07 | 0.06 | 0.06 | 0.8 | 0.81 |
| 2010 | 0.82 | 0.73 | 0.1 | 0.11 | 0.09 | 0.11 | 0.05 | 0.06 | 0.06 | 0.05 | 0.81 | 0.83 |
| 2011 | 0.83 | 0.74 | 0.09 | 0.11 | 0.09 | 0.1 | 0.06 | 0.06 | 0.06 | 0.05 | 0.82 | 0.84 |

| | | | | | | | | | | | |
|---|---|---|---|---|---|---|---|---|---|---|---|
| 2012 | 0.81 | 0.72 | 0.12 | 0.13 | 0.09 | 0.12 | 0.06 | 0.05 | 0.05 | 0.05 | 0.81 | 0.82 |
| 2013 | 0.82 | 0.73 | 0.09 | 0.12 | 0.09 | 0.13 | 0.06 | 0.07 | 0.05 | 0.07 | 0.8 | 0.82 |
| 2014 | 0.85 | 0.74 | 0.09 | 0.11 | 0.08 | 0.09 | 0.06 | 0.08 | 0.05 | 0.06 | 0.83 | 0.85 |
| 2015 | 0.79 | 0.69 | 0.12 | 0.14 | 0.1 | 0.12 | 0.07 | 0.09 | 0.07 | 0.07 | 0.79 | 0.81 |

Note: NSE is from the evaluation with the time series of average 0.25º pixels while the other five metrics are from the evaluation with 1 km disaggregated values.

3. L143 please check grammar and explain why regular DEM wasn't used

Response: Earlier studies [1] found that topographic position index (TPI) correlated best with surface variables such as snow depth and soil moisture amongst several other terrain-derived indices, with the strength of the correlation varying as a function of elevation range and topographic complexity in the designated neighborhood. Larger neighborhoods are more likely to reveal larger-scale terrain features such as valleys and ridges, while smaller neighborhoods more likely identify local depressions/fissures [2]. Meanwhile, our analysis observes that TPI has the higher Pearson's correlation with soil moisture in comparison with DEM (0.317 versus 0.288 in significance percentage, and 0.172 versus 0.153 in significance percentage), which encourages our work for using TPI rather than the DEM. In the new version, we will add the related context.

[1] Cristea N C, Breckheimer I, Raleigh M S, et al. An evaluation of terrain-based downscaling of fractional snow covered area data sets based on LiDAR-derived snow data and orthoimagery. Water Resources Research, 2017, 53(8): 6802-6820.
[2] Shaw T E, Gascoin S, Mendoza P A, et al. Snow depth patterns in a high mountain Andean catchment from satellite optical tristereoscopic remote sensing. Water Resources Research, 2020, 56(2): e2019WR024880.

4. L335 (important comment) please explain the reasoning behind the separation percentage of 90 10 training testing and how this subset was extracted. Is it random undefined subset or is a certain defined period was extracted and why.

Response: The artificial gaps were created to mitigate potential bias in model performance for any particular gap sequence. Model performance was tested by 10-fold cross validation (CV). The gap-filling accuracy was evaluated by a spatiotemporal 10-fold CV, i.e., randomly holding out 10% of pixels, training the model with the remaining 90% of the pixels, making evaluating on the held-out stations, and repeating this process 10 times. Previous studies generally implemented regular spatial or temporal 10-fold CV by randomly choosing 10% subset of each band or 10% period of each grid each time. However, a much stricter spatial and temporal CV procedure [1, 2] was selected in this study by rearranging the

pixels of during all studied period into one time series and then dropping 10% samples each time (leave-one-year-out-cross-validation) to test the capacity of gap-filling soil moisture. According to the reviewer's suggestion, we will revise the related context in this version.

[1] Meng X, Liu C, Zhang L, et al. Estimating PM2. 5 concentrations in Northeastern China with full spatiotemporal coverage, 2005–2016. Remote sensing of environment, 2021, 253: 112203.
[2] Zhang D, Du L, Wang W, et al. A machine learning model to estimate ambient PM2. 5 concentrations in industrialized highveld region of South Africa. Remote Sensing of Environment, 2021, 266: 112713.

5. Figure 13: please reword the description of panel (a) regarding the text in the figure, does that describe the decrease in performance as a percentage of the original value?
Response: This denotes the relative percentage of the decreased accuracy in comparison to the baseline accuracy using all seven variables. We will revise the related context in the new version.

6. Figure 15 please explain in writing the effect of the significance level on the accuracy of conclusions and interpretation of the results.
Response: As shown in Fig. 15(d)-(f), the difference in valid SM values participating in trend analysis causes a disparity in calculating SM trend, i.e., bringing a lower SM trend in most wet regions but an increased SM trend in some dry regions when gap-filled values are introduced. As a result of the missing satellite soil moisture retrievals, the 8-days SM tends to be overestimated. In particular, most of the regions owing significant trend demonstrate a lower trend when comparison to the original values. Meanwhile, the confidence level of SM trend is converted from a significance level to non-significance level for a considerable fractional of the grids. This is more pronounced in wet regions such as northeast, northwest and southwest parts of China, which is sensitive to monsoon precipitation and ice melting. Our results are corroborated by earlier studies that revealed an overestimation in trend of missing AOD [1] and albedo [2] when the cloudy conditions prevented satellite retrievals. In general, the variations in SM trend are related to the climate variables (e.g., precipitation) changes and land management activities (Li et al., 2018). According to the reviewer's suggestion, we will revise the related context in this version.

[1] Zhang R, Di B, Luo Y, et al. A nonparametric approach to filling gaps in satellite-retrieved aerosol optical depth for estimating ambient PM2. 5 levels. Environmental Pollution, 2018, 243: 998-1007.
[2] Gunnarsson A, Gardarsson S M, Pálsson F, et al. Annual and inter-annual variability and trends of albedo of Icelandic glaciers. The Cryosphere, 2021, 15(2): 547-570.

7. Figure 16 is difficult to interpret, please make sure to have a proper legend that makes the figures self-explanatory, especially in the lower panels. Reading the figure caption to understand symbols adds difficulty to the interpretation.

Response: Thanks for pointing out this issue. The original Figure 16 and the related caption will be adjusted in the new version.

[Figure]

Figure 14: (a) shows the temporal patterns of raw CCI SM regarding different climate regions during 2005-2015, and (b) shows the temporal patterns of gap-filled CCI SM regarding different climate regions. The shaded area in (a) and (b) denotes ±1 standard error. (c) and (d) The scatter plot of 1-km CCI SM-derived trends against in situ measures during 2005-2014, and (c) shows the trends under significance level (P<0.05), while (d) shows all the trends. (e) and (f) The scatter plot of 1-km gap-filled SM-derived trends against in situ measures during 2005-2014, and (e) shows the trends under significance level, while (f) shows all the trends.

References

Li, B., Liang, S., Liu, X., Ma, H., Chen, Y., Liang, T., and He, T.: Estimation of all-sky 1 km land surface temperature over the conterminous United States, Remote Sensing of Environment, 266, 112707, https://doi.org/10.1016/j.rse.2021.112707, 2021.

Li, Q., Wang, Z., Shangguan, W., Li, L., Yao, Y., and Yu, F.: Improved daily SMAP satellite soil moisture prediction over China using deep learning model with transfer learning, Journal of Hydrology, 600, 126698, https://doi.org/10.1016/j.jhydrol.2021.126698, 2021b.

Li, Q., Li, Z., Shangguan, W., Wang, X., Li, L., and Yu, F.: Improving soil moisture prediction using a novel encoder-decoder model with residual learning, Computers and Electronics in Agriculture, 195, 106816, https://doi.org/10.1016/j.compag.2022.106816, 2022b.

Li, L., Dai, Y., Shangguan, W., Wei, N., Wei, Z., and Gupta, S.: Multistep Forecasting of Soil Moisture Using Spatiotemporal Deep Encoder–Decoder Networks, Journal of Hydrometeorology, 23, 337-350, 10.1175/jhm-d-21-0131.1, 2022.

Li, Y., Piao, S., Li, L. Z. X., Chen, A., Wang, X., Ciais, P., Huang, L., Lian, X., Peng, S., Zeng, Z., Wang, K., and Zhou, L.: Divergent hydrological response to large-scale afforestation and vegetation greening in China, Science Advances, 4, eaar4182, doi:10.1126/sciadv.aar4182, 2018.

Long, D., Bai, L., Yan, L., Zhang, C., Yang, W., Lei, H., Quan, J., Meng, X., and Shi, C.: Generation of spatially complete and daily continuous surface soil moisture of high spatial resolution, Remote Sensing of Environment, 233, 111364, https://doi.org/10.1016/j.rse.2019.111364, 2019.

Shangguan, W., Hengl, T., Mendes de Jesus, J., Yuan, H., and Dai, Y.: Mapping the global depth to bedrock for land surface modeling, Journal of Advances in Modeling Earth Systems, 9, 65-88, https://doi.org/10.1002/2016MS000686, 2017.

---

## Author Response (AR1)

**Response to #1**

The study presents a method for gap-filling ESA CCI soil moisture data. For filling the gaps, the approach utilizes information from the spatiotemporal domain around the missing value as well as from other explanatory variables. This is a timely contribution to the ever-growing field of gap-filling and data fusion in Earth system science. The study benchmarks the method over China, which covers a large variety of different climate zones and topography, making it a suitable test bed. Particularly with the severity of missing values in microwave remote sensing soil moisture retrievals, it is vital to use as much information as possible, both from the spatial and temporal domain as well as from other available observations. Therefore, the proposed method seems promising to be able to do the task. Furthermore, comparing the gap filled values to in-situ observations gives an independent insight into gap filling performance.

However, the poor use of language and grammar hinder understanding of the methods and results in many cases. Additionally, the structure of the results and methods sections are a bit unclear and seem arbitrary, such that following the storyline of the paper is difficult. Finally, the purpose, aim and implementation of some of the methodological choices and evaluations in the result section are unclear and should either be replaced or clarified.

I therefore think the paper requires major revisions. Please see below my general and specific comments. Response: Thanks for your valuable time and constructive comments. We tried our best to incorporate them in the new version. Especially, the English language usage in this version has been improved deeply by one fluent English researcher that is suggested by the commercial corporation. The detailed modifications can be found in the highlighted text throughout the manuscript.

**GENERAL COMMENTS**

1. The study needs restructuring especially in the methods and results section. It is hard to follow the methods and results section, since they are structured very differently. Please align Figure 2 with the structure of Section 3, such that it is easy for the reader to follow which part of the model is described. For example, the subsections could have the same header names as the boxes in Figure 2. Also, it is not clear why only parts of Figure 2 are described in the subsections. Furthermore within the results, sometimes two different results are explained in one subsection (e.g. Section 4.2. compares to in-situ measurements and to the "cross-validation", Section 4.3 compares to literature and to other employed methods). I suggest making individual sections for individual results, or clarify why the results are thrown together in one Section like this.

Response: Thanks for pointing out this issue. In the revised version, the Figure 2 and the related context have been rewritten to make it align with the structure of section 3. Meanwhile, the Section 3 and 4 is

reorganized, in particular, the subsection in these two sections are rearranged to make it clear and readable.

Figure 2: The schematic of overall procedure. The red text denotes the core procedures conducted in the proposed model, which will be described in the following sections.

2. In general within the result section, the results are described with generic terms like "good coindicence", "capacity for reconstruction", "high accuracies", "strong variation delineating capacity". Whilst those all are incorrect English terms, they also do not go into detail or describe the results well. While it is important to give statistical evidence of the algorithm performance, it is of equal importance to discuss the difference between the statistical measures and possible reasons for it, as well as evaluating the physical plausibility and coherency of the gap filled values. It would highly benefit the study if the results were described more concisely, more descriptive words would be used, and the results would be discussed, taken into context and described better. A good example of a good explanation of results is for example found in L433-436. Please more of those. Furthermore, it would be beneficial to add at least one plot that evaluates the physical plausibility of the gap-filled values for a possible application of the gap filled dataset. As an example, since soil moisture values are often important when analysing droughts, it would be interesting to see whether the gap-filling method is able to not only work in the mean of the values, which all the statistical metrics aim at, but also see whether the extreme values are gap-filled well. This could be done by showing some maps of a known drought event over China, possibly compare with the in-situ measurements and the original, gappy CCI SM data.

Response: Thanks for the reviewer's suggestion. In the new version, the result section has been revised and more descriptive words are added to make it more convinced and concise, e.g., regarding the applicability of the selected explanatory factors and the merits of RF model in gap-filling. The detailed modifications can be found in the highlighted text throughout the manuscript.

According to the reviewer's suggestions, we pay particular attention to the gap-filling SMs under extremely dry conditions to reveal the physical plausibility of gap-filled soil moisture. The extreme drought is defined based on meteorological condition, i.e., the Palmer Drought Severity Index (PDSI) is less than -2 over 8 consecutive months or longer (Fig. S2). Specifically, 1) we add the accuracy evaluation in Fig. 7 and 8 focusing on the drought regions, and find the accuracy of the gap-filling products tend to be diminished by drought conditions, but the impact is limited; 2) we label the drought regions in Fig. 9 to make it more readable, and we observe the accuracies are lower over the regions experienced drought due to perturbations of the soil water content, but without noticeably poor performances; and 3) we analyze two typical drought regions by comparing the difference source SM (Fig. S4). The focused analyses illustrate the consistency of the gap-filling SM with the in-situ measurements and the original SM under extremely dry conditions, illustrating the physical plausibility of the gap-filled values for specific application. The detailed modifications can be found in the highlighted text in the revised version.

---

## Referee Report (RR1)

Major comments have been addressed in the review. I think the paper therefore requires minor revisions. Please see below my general and specific comments.

GENERAL COMMENTS

I would like to again, add as a cautionary note, that comparing "raw" and "gapfilled" values and exploring their difference is not related to the merit of the gap-fill, due to the complex missingness pattern of the soil moisture data. I would discuss this more prominently in Section 4.1 and Section 4.5. The differences between "raw" and "gap filled" are however very interesting, and likely related to the pattern of missing values that is underlying. I find this very interesting to discuss and would encourage you to write a bit more about it in those Sections. In the abstract and the introduction you cite satellite coverage and radio-frequency interference as a mechanism for the missing values in ESA CCI SM. Note however that according to Dorigo et al, 2017, those are not the only one and quite possibly not the major contributors to missing data. In this citation, they also note that snow-covered pixels are removed since the water in the snow is measured by the satellite, not the soil moisture, in these situations. Furthermore, regions with high vegetation cover are set to missing in post processing, since there the signal cannot penetrate through the vegetation and reports, respectively, only the water stored in the above-ground vegetation. I haven't looked into how those two missingness mechanisms play a role in your study area, but it could be that the snow-cover problem is responsible for many missing values in winter (e.g. Fig 5a) and the overall wetter gap-filled values in high latitude regions in Figure 13 and Figure 5b,c for winter. Or, it could be due to vegetation cover. It might be interesting to investigate, for example by plotting Figure 13 separately for each month, to see where the "wet" signal in the gapfill is coming from and which missingness mechanism it can be explained by, if any.

Unfortunately, I still can't wrap my head around the applied variable correction. I understand now that this is applied to the ERA SM data before using it for the gap-filling. I agree that a bias is very likely to exist between ESA SM and ERA SM and needs to be corrected before using traditional machine learning methods. But I don't yet understand what the difference between Equations (3) is and standardised anomalies. When I inspect the equations, I understand that the result is the mean of the ESA SM plus the anomalies of the ERA SM, scaled with the ratio of their standard deviations. To me, this is an equal procedure as using standardised anomalies for both ERA SM and ESA SM. However, both is fine, so no need to change the variable correction. However, also note that for the Random Forest method, scaled/standardised variables are not necessary, since the scale of a feature (input variable) does not depend on its importance after training. (E.g see https://datascience.stackexchange.com/questions/62031/normalize-standardize-in-a-random-forest https://stackoverflow.com/questions/8961586/do-i-need-to-normalize-or-scale-data-for-randomforest-r-package; I could not find a literature source for this unfortunately but it derives logically from the RF function). So you might do additional work here that is, after all, not necessary, if I understood everything correctly.

As far as I understood, the training period of the Random Forest, described in Section 3.2.1, is on the years 2003-2008, and therefore overlaps with the Long term extension, described in Section 4.5., that runs from 2005-2015. Since using the same data for training and predicting with a Random Forest likely leads to overestimation on the merit of the Random Forest, I suggest adding a cautionary

note to that effect in the text.

SPECIFIC COMMENTS
L12 and L58+59: radio-frequency interference is mentioned as a major reason for gaps in the ESA CCI SM data. However, many values in the ESA CCI SM are also missing because of snow cover, frozen soil, or dense vegetation (Dorigo et al, 2017). Since those mechanisms are relevant for the interpretation of the results (see general comments above), I think they should be included here.

L29: missing comma "soil moisture (SM),"

L39 optionally add Dorigo et al, 2021

L84: sentence a bit unclear. Is "compared" meant instead of "focused"?

L65: Bessenbacher et al, 2022 useful citation here

L65, L197, L357, L523: "strong capacity/capability" leads in my opinion to convoluted sentences and is not very clear. Try to not use as much.

L67: no brackets needed around citation

L95: missing space character before Zhang et al

L128ff: add in brackets behind (i), (ii), (iii) where they can be found in Fig. 2. Example: "(i) use a regression (…) bias between them (Fig2 Part 1 red text);"

L186: possible typo "MODSI"

L207: remove "reliable"

L207: presented -> present

L240: if available, it would be really interesting to know what the computational efficiency of the presented gap-filling algorithm was, i.e. how long it roughly ran on how many CPUs / which machine. If possible, would be nice to include this information!

L246: missing space character before citation

L259: is 2003-2008 the training window or the cross-validation window for the RF parameters? Or both? Please clarify

L262: remove "best"

L291: Do you perhaps mean Eq (1)?

L314: is suppose to -> can

Figure5: optional suggestion: put Figure S3 here and move Figure5 (b, d) into S3.

Makes more sense to me, but just optional.

L375-376: severe contamination in winter season is likely related to snow cover in ESA CCI SM, list as reason

L279ff: You state that in the warm season, more missing pixels are reconstructed than in the cold season. You argue this is because surface coupling is larger in summer. This might be true, but how does the algorithm know this? I thought it only leaves gaps unfilled when there are not sufficient neighbouring points available to run the Random Forest. How does this related to the strength of the surface coupling? Please clarify.

L382: Yes they tend to produce "bias", however, this is not a problem itself. If the pixels are missing not at random, I would expect bias between "raw" and "gap filled", because the gap filled data might e.g. fill especially pixels under dense vegetation, which are systematically wetter than pixels with less dense vegetation. Therefore, this "bias" is not a bad thing, I would be worried if there wasn't any.

L410: consider moving this part of 4.2 into a separate (sub)section.

L412: "better" compared to what? As R2 and RMSE are not comparable, because of different units.

L419: unnecessary space character after bracket

L426: "some grids" unclear. Some regions?

L522: add year 2009 to the table for easy comparison

L525ff: Add that the gap-filled product is drier overall than the "raw" is consistent with the findings in Figure 5.

L527: "overestimated": since we don't know the ground truth of missing values, I would not use words like "overestimated"

L531: important finding: trends could be overestimated in satellite SM due to missingness.

L535f: How does missing values in AOD and missing values in albedo compare to missing values in SM? Since the missing values mechanisms (below clouds) are different from soil moisture (vegetation, snow, interference), why would we expect the effect of gap-filling to be similarly, e.g. reducing trends? This might be related to the next sentence in the text, but unfortunately I don't understand this sentence. Please clarify.

L549: Unfortunately I don't understand Figure S7. Is that like Figure14, but disaggregated into trends and seasonal cycle? Then, trends are barely visible and cannot be compared like this.

L554: "Overall…" unclear statement, please clarify.

Figure14a,b: These values are hard to compare. Idea: combine a and b into 1 plot and mark "raw" and "gap-filled" with different colours. Same for S7

REFERENCES
Dorigo et al, 2017: ESA CCI Soil Moisture for improved Earth system understanding: State-of-the art and future directions. Remote Sensing of Environment. Volume 203, Pages 185-215, https://doi.org/10.1016/j.rse.2017.07.001

Dorigo et al, 2021: The International Soil Moisture Network: serving Earth system science for over a decade. Hydrol. Earth Syst. Sci. 10.5194/hess-25-5749-2021

Bessenbacher et al, 2022: CLIMFILL v0.9: A Framework for Intelligently Gap filling Earth Observations, GMD, https://doi.org/10.5194/gmd-15-4569-2022

---

## Author Response (AR2)

**Reply to reviewer 1**

Major comments have been addressed in the review. I think the paper therefore requires minor revisions. Please see below my general and specific comments.

GENERAL COMMENTS

I would like to again, add as a cautionary note, that comparing "raw" and "gapfilled" values and exploring their difference is not related to the merit of the gap-fill, due to the complex missingness pattern of the soil moisture data. I would discuss this more prominently in Section 4.1 and Section 4.5. The differences between "raw" and "gap filled" are however very interesting, and likely related to the pattern of missing values that is underlying. I find this very interesting to discuss and would encourage you to write a bit more about it in those Sections. In the abstract and the introduction you cite satellite coverage and radio-frequency interference as a mechanism for the missing values in ESA CCI SM. Note however that according to Dorigo et al, 2017, those are not the only one and quite possibly not the major contributors to missing data. In this citation, they also note that snow-covered pixels are removed since the water in the snow is measured by the satellite, not the soil moisture, in these situations. Furthermore, regions with high vegetation cover are set to missing in post processing, since there the signal cannot penetrate through the vegetation and reports, respectively, only the water stored in the above-ground vegetation. I haven't looked into how those two missingness mechanisms play a role in your study area, but it could be that the snow-cover problem is responsible for many missing values in winter (e.g. Fig 5a) and the overall wetter gap-filled values in high latitude regions in Figure 13 and Figure 5b,c for winter. Or, it could be due to vegetation cover. It might be interesting to investigate, for example by plotting Figure 13 separately for each month, to see where the "wet" signal in the gapfill is coming from and which missingness mechanism it can be explained by, if any. Unfortunately, I still can't wrap my head around the applied variable correction. I understand now that this is applied to the ERA SM data before using it for the gapfilling. I agree that a bias is very likely to exist between ESA SM and ERA SM and needs to be corrected before using traditional machine learning methods. But I don't yet understand what the difference between Equations (3) is and standardized anomalies. When I inspect the equations, I understand that the result is the mean of the ESA SM plus the anomalies of the ERA SM, scaled with the ratio of their standard deviations. To me, this is an equal procedure as using standardized anomalies for both ERA SM and ESA SM. However, both is fine, so no need to change the variable correction. However, also note that for the Random Forest method, scaled/standardised variables are not necessary, since the scale of a feature (input variable) does not depend on its importance after training. (E.g see https://datascience.stackexchange.com/questions/62031/normalize-standardize-in-arandom- forest https://stackoverflow.com/questions/8961586/do-i-need-to-normalizeor-scale-data-for-randomforest-r-package; I could not find a literature source for this unfortunately but it derives logically from the RF

function). So you might do additional work here that is, after all, not necessary, if I understood everything correctly. As far as I understood, the training period of the Random Forest, described in Section 3.2.1, is on the years 2003-2008, and therefore overlaps with the Long term extension, described in Section 4.5., that runs from 2005-2015. Since using the same data for training and predicting with a Random Forest likely leads to overestimation on the merit of the Random Forest, I suggest adding a cautionary note to that effect in the text.

Response: Thanks for your valuable time and constructive comments. We tried our best to incorporate them in the new version. The detailed modifications can be found in the highlighted text throughout the manuscript.

SPECIFIC COMMENTS

1. L12 and L58+59: radio-frequency interference is mentioned as a major reason for gaps in the ESA CCI SM data. However, many values in the ESA CCI SM are also missing because of snow cover, frozen soil, or dense vegetation (Dorigo et al, 2017). Since those mechanisms are relevant for the interpretation of the results (see general comments above), I think they should be included here.

Response: Thanks for pointing out this issue. We have revised this in the new version.

2. L29: missing comma "soil moisture (SM),"

Response: We have revised this in the new version.

3. L39 optionally add Dorigo et al, 2021

Response: We have added this in the new version.

4. L84: sentence a bit unclear. Is "compared" meant instead of "focused"?

Response: Thanks for pointing out this issue. We have deleted this in the new version.

5. L65: Bessenbacher et al, 2022 useful citation here

Response: We have added this in the new version.

6. L65, L197, L357, L523: "strong capacity/capability" leads in my opinion to convoluted sentences and is not very clear. Try to not use as much.

Response: Thanks for pointing out this issue. This has been revised through the manuscript.

7. L67: no brackets needed around citation

Response: We have revised this in the new version.

8. L95: missing space character before Zhang et al

Response: We have revised this in the new version.

9. L128ff: add in brackets behind (i), (ii), (iii) where they can be found in Fig. 2. Example: "(i) use a regression (…) bias between them (Fig2 Part 1 red text);"

Response: Thanks for pointing out this issue. We have revised this in the new version.

10. L186: possible typo "MODSI"

Response: We have revised this in the new version.

11. L207: remove "reliable"

Response: According to the reviewer's suggestion, this has been revised in the new version.

12. L207: presented -> present

Response: We have revised this in the new version.

13. L240: if available, it would be really interesting to know what the computational efficiency of the presented gap-filling algorithm was, i.e. how long it roughly ran on how many CPUs / which machine. If possible, would be nice to include this information!

Response: In this study, the random forest model for gap-filling SM is implemented in 2 CPU cores. It approximately takes 12 hours in gap-filling 1-year CCI dataset over China.

14. L246: missing space character before citation

Response: We have revised this in the new version.

15. L259: is 2003-2008 the training window or the cross-validation window for the RF parameters? Or both? Please clarify

Response: In our study, the Bayesian optimization process is implemented by using the dataset of 2003—2008 as the cross-validation window. According to the reviewer's suggestion, we have revised this in the new version.

16. L262: remove "best"

Response: We have revised this in the new version.

17. L291: Do you perhaps mean Eq (1)?

Response: Thanks for pointing out this issue. We have revised this in the new version.

18. L314: is suppose to -> can

Response: We have revised this in the new version.

19. L375-376: severe contamination in winter season is likely related to snow cover in ESA CCI SM, list as reason

Response: We have added this in the new version.

20. L379ff: You state that in the warm season, more missing pixels are reconstructed than in the cold season. You argue this is because surface coupling is larger in summer. This might be true, but how does the algorithm know this? I thought it only leaves gaps unfilled when there are not sufficient neighbouring points available to run the Random Forest. How does this relate to the strength of the surface coupling? Please clarify.

Response: This is only one speculation. Although this has been reported by previous studies, we can not provide direct evidence. According to the reviewer's suggestion, we have revised this in the new version.

21. L382: Yes they tend to produce "bias", however, this is not a problem itself. If the pixels are missing not at random, I would expect bias between "raw" and "gap filled", because the gap filled data might e.g. fill especially pixels under dense vegetation, which are systematically wetter than pixels with less dense vegetation. Therefore, this "bias" is not a bad thing, I would be worried if there wasn't any.

Response: Thanks for the reviewer's suggestion. We have revised this in the new version.

22. L410: consider moving this part of 4.2 into a separate (sub)section.

Response: Thanks for pointing out this issue. We have revised this in the new version.

23. L412: "better" compared to what? As R2 and RMSE are not comparable, because of different units.

Response: Thanks for pointing out this issue. We have revised this in the new version.

24. L419: unnecessary space character after bracket

Response: We have revised this in the new version.

25. L426: "some grids" unclear. Some regions?

Response: We have revised this in the new version.

26. L522: add year 2009 to the table for easy comparison

Response: Thanks for pointing out this issue. We have added this in the new version.

27. L525ff: Add that the gap-filled product is drier overall than the "raw" is consistent with the findings in Figure 5.

Response: According to the reviewer's suggestion, we have revised this in the new version.

28. L527: "overestimated": since we don't know the ground truth of missing values, I would not use words like "overestimated"

Response: We have deleted this in the new version.

29. L531: important finding: trends could be overestimated in satellite SM due to missingness.

Response: Thanks for pointing out this issue. We have added this in the new version.

30. L535f: How does missing values in AOD and missing values in albedo compare to missing values in SM? Since the missing values mechanisms (below clouds) are different from soil moisture (vegetation, snow, interference), why would we expect the effect of gap-filling to be similarly, e.g. reducing trends? This might be related to the next sentence in the text, but unfortunately I don't understand this sentence. Please clarify.

Response: This merely supports the missing values in satellite data could result in bias of trend. We can not provide one mechanism among difference variables. According to the reviewer's suggestion, we have removed this to the next paragraph.

31. L549: Unfortunately I don't understand Figure S7. Is that like Figure14, but disaggregated into trends and seasonal cycle? Then, trends are barely visible and cannot be compared like this.

Response: Thanks for pointing out this issue. In the new version, we have deleted this Figure.

32. L554: "Overall…" unclear statement, please clarify.

Response: We have deleted this in the new version.

33. Figure14a, b: These values are hard to compare. Idea: combine a and b into 1 plot and mark "raw" and "gap-filled" with different colors. Same for S7

Response: Thanks for the reviewer's suggestion. In the new version, we have rewritten the Fig. 14.

[Figure]

**Figure 14: (a) shows the temporal patterns of raw and gap-filled CCI SM regarding different climate regions during 2005-2015. The shaded area in (a) denotes ±1 standard error. (b) and (c) Scatter plot of 1-km CCI SM-derived trends against in situ measures during 2005-2014, and (b) shows the trends under significance level, while (c) shows all the trends. (d) and (e) Scatter plot of 1-km gap-filled SM-derived trends against in situ measures during 2005-2014, and (d) shows the trends under significance level, while (e) shows all the trends.**

REFERENCES

Dorigo et al, 2017: ESA CCI Soil Moisture for improved Earth system understanding: State-of-the art and future directions. Remote Sensing of Environment. Volume 203, Pages 185-215, https://doi.org/10.1016/j.rse.2017.07.001

Dorigo et al, 2021: The International Soil Moisture Network: serving Earth system science for over a decade. Hydrol. Earth Syst. Sci. 10.5194/hess-25-5749-2021

Bessenbacher et al, 2022: CLIMFILL v0.9: A Framework for Intelligently Gap filling Earth Observations, GMD, https://doi.org/10.5194/gmd-15-4569-2022

**Reply to reviewer 2**

The authors satisfactorily responded to the review comments. My comments are resolved except for Figure 14. I would suggest editing Figure 14 by giving more room to panels (d) and (f). This can be done by having panel c,d in a row and e,f in another row below. Maybe even put (d) through (f) in a different figure. I don't think it's appropriate to have content that cannot be read in 100% zoom level.

Response: Thanks for the reviewer's suggestion. According to the reviewers and editor's suggestion, we have rewritten the Fig. 14 to make it more readable and visible.

[Figure]

**Figure 14: (a) shows the temporal patterns of raw and gap-filled CCI SM regarding different climate regions during 2005-2015. The shaded area in (a) denotes ±1 standard error. (b) and (c) Scatter plot of 1-km CCI SM-derived trends against in situ measures during 2005-2014, and (b) shows the trends under significance level, while (c) shows all the trends. (d) and (e) Scatter plot of 1-km gap-filled SM-derived trends against in situ measures during 2005-2014, and (d) shows the trends under significance level, while (e) shows all the trends.**